# Value added transformation of ubiquitous substrates into highly efficient and flexible electrodes for water splitting

Atharva Sahasrabudhe[1], Harsha Dixit[1], Rahul Majee[1] & Sayan Bhattacharyya [1]

Herein, we present an innovative approach for transforming commonly available cellulose paper into a flexible and catalytic current collector for overall water splitting. A solution processed soak-and-coat method of electroless plating was used to render a piece of paper conducting by conformably depositing metallic nickel nanoparticles, while still retaining the open macroporous framework. Proof-of-concept paper-electrodes are realized by modifying nickel-paper current collector with model electrocatalysts nickel-iron oxyhydroxide and nickel-molybdenum bimetallic alloy through electrodeposition route. The paper-electrodes demonstrate exceptional activities towards oxygen evolution reaction and hydrogen evolution reaction, requiring overpotentials of 240 and 32 mV at 50 and $-10\,\mathrm{mA\,cm^{-2}}$, respectively, even as they endure extreme mechanical stress. The generality of this approach is demonstrated by fabricating similar electrodes on cotton fabric, which also show high activity. Finally, a two-electrode paper-electrolyzer is constructed which can split water with an efficiency of 98.01%, and exhibits robust stability for more than 200 h.

[1] Department of Chemical Sciences, and Centre for Advanced Functional Materials, Indian Institute of Science Education and Research (IISER) Kolkata, Mohanpur 741246, India. Correspondence and requests for materials should be addressed to S.B. (email: sayanb@iiserkol.ac.in)

Electricity driven splitting of water into its constituents hydrogen and oxygen, is one of the attractive routes in the conversion and storage of different renewable energy sources into useable fuel forms[1,2]. To improve the overall electrolysis efficiency, rational design of the electrode (electrocatalyst + current collector) architecture is crucial[3,4]. Electrodes free of polymeric binders possess increased density of exposed active sites that render faster mass-transport[5,6]. The existing approaches with carbon-based and metal-foam/foil based substrates suffer from drawbacks such as high cost procedures and oxidation sensitivity of carbon supports, unsuitability for large scale fabrication, structural fragility of annealed metal-foam and low conductivity at high temperatures[7–10]. Transformation of mechanically robust substrates into perfectly conducting yet flexible current collectors, without compromising their overall catalytic activity constitutes an exciting idea to overcome the aforementioned bottle-necks. Moreover, it is highly desirable that such transformations use ubiquitous precursors, be economical and solution processable for them to be scalable and of commercial value.

In this regard, the recent developments in portable and wearable electronics have fueled research activity in the area of bendable and foldable current collectors[11–14]. One such breakthrough was provided by the Whitesides group in 2007 wherein they introduced common cellulose paper as a flexible and cheap substrate to build micro-fluidic paper-based analytical devices[15]. Since then paper-electronics[16] has enabled a myriad of different applications where chemically/physically treated paper plays the role of a flexible current collector, for example "pencil-on-paper" piezoresistive sensors reported by Kang[17] and pencil-drawn strain gauges/chemiresistors on paper demonstrated by Huang group[18]. Similarly, Cui's group utilized SWCNT coated cellulose paper for building solid state supercapacitors[19], while paper-based bacteria powered batteries were realized by Choi et al.[20]. While significant research efforts have been directed towards integrating paper-electronics with sensors and energy storage devices, only limited advancements have been made with developing energy conversion devices out of conductive paper-based current collectors. For example paper-based nanogenerators as a power-source for mechanical energy conversion were reported by Wang's group[21], while Rojas et al. successfully demonstrated thermoelectric generator out of paper substrates for harnessing heat energy[22]. However, there have been no prior attempts at transforming commonly found piece of paper into a flexible and catalytic electrode for converting electrical energy into chemical fuels as in overall splitting of water.

From the view point of electrocatalysis, paper is an ideal substrate for growing active catalyst layers owing to the porous microstructure and abundance of co-ordinating hydroxyl and epoxy functional groups on the cellulose micro-fibrils which together can result in strong binding and faster mass-transport kinetics, apart from the added functionality of flexibility. Moreover, flexible current collectors can render reduced internal strain during repeated electrochemical processes and have potential applications for wearable energy storage devices and bioelectronic sensors[23,24]. To fully exploit these useful properties inherent in a paper substrate, we make use of the solution processed "dip-and-coat" method of electroless metal plating to deposit non-noble $Ni^0$ nanoparticles (NPs) on cellulose paper; thereby converting it into a porous, conducting and flexible current collector. The nickel coated conducting paper (abbreviated as Ni-P henceforth) is further used as flexible current collector for fabricating a paper-anode and a paper-cathode by electrodepositing the representative electrocatalyst systems of $Ni_xFe_{1-x}OOH$ layered double oxyhydroxide and $Ni_4Mo$ bimetallic alloy for the oxygen and hydrogen evolution half reactions (OER and HER), respectively. The choice of these particular electrocatalysts as model systems out of several other high performing counterparts is not arbitrary and is based upon rationalized principles. The flexible paper-electrodes demonstrate remarkable efficiencies towards water splitting even after enduring high mechanical stress and harsh alkaline conditions, with prolonged stabilities and excellent reproducibility, which outperforms the traditional combination of noble benchmarks of Pt and $IrO_x$. Full water splitting with ~98% electrolysis efficiency is achieved at just 1.51 V to generate a current density of $10\,mA\,cm^{-2}$ in a two-electrode proof-of-concept "paper-electrolyzer". Impressively the paper-electrodes exhibit robust stability of more than 200 h of continuous operation in 1 M KOH and at least 12 h in 10 M KOH. Moreover, the general applicability of this methodology is also demonstrated by transforming commonly available cotton fabric into highly efficient OER and HER electrodes in a similar fashion.

## Results

**Unmodified Ni-P electrodes**. The design and fabrication strategy of different paper electrodes is represented in Fig. 1. Common

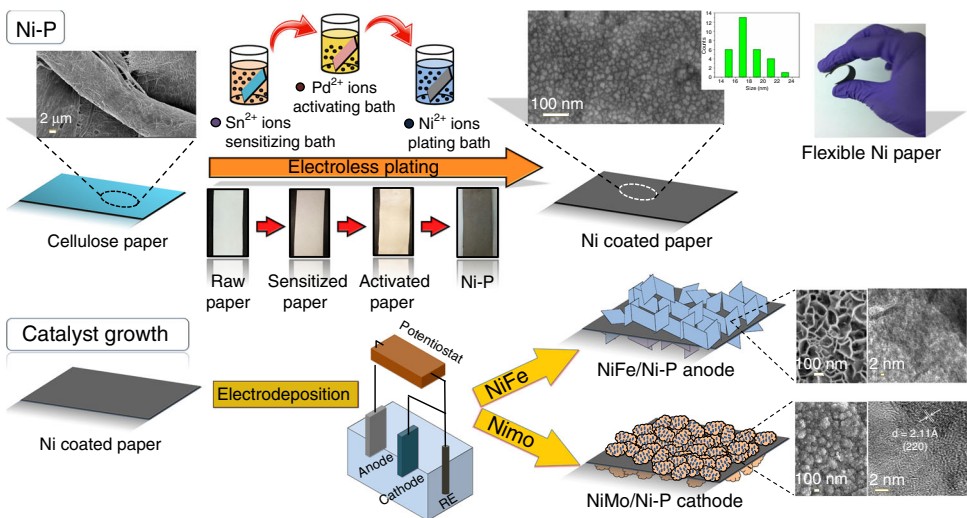

**Fig. 1** Schematic illustration of the fabrication process. Digital images, field-emission scanning electron microscopy and transmission electron microscopy images depicting various steps in the formation process of Ni-P, NiFe/Ni-P, and NiMo/Ni-P electrodes

paper is composed of polymeric cellulose strands which have abundant –OH functional groups on the surface that facilitate the strong anchoring of different precursors that are used during electroless plating of metallic $Ni^0$. Electroless plating is a well-established process which involves reduction of metal ions from an aqueous bath with a reducing agent (e.g., $NaH_2PO_2$) and a catalyst (e.g., $Pd^0$)[25–28]. While the reducing agent is simply dissolved in the plating bath, the catalyst is deposited onto the substrate to be coated through a redox reaction involving $Sn^{2+}$ ions. This two-step process is summarized in the Eqs. 1 and 2:

$$Sn^{2+} + Pd^{2+} \rightarrow Sn^{4+} + Pd^0, \tag{1}$$

$$Ni^{2+} + H_2PO_2^- + H_2O \xrightarrow{Pd\ Catalyst} Ni^0 + H_2PO_3^- + 2H^+. \tag{2}$$

Sensitization of the cellulose paper by $Sn^{2+}$ results in pale yellow coloration while subsequent activation by $Pd^{2+}$ ions produces a light brown texture which is retained even after multiple washing steps, indicating strong binding of the ions. The anchored $Pd^0$, from reduction of $Pd^{2+}$ by $Sn^{2+}$, further catalyze the reduction of $Ni^{2+}$ to $Ni^0$ NPs in presence of $NaH_2PO_2$ as the reducing agent, and the color of the paper simultaneously changes to ash-gray, indicating successful formation of metallic $Ni^0$, albeit with leftover traces of ~1.2% Pd and ~2.6% Sn trapped within the bulk of the porous paper (Supplementary Fig. 1a). The self-catalyzed and closely packed ~18 nm Ni NP coating does not alter the porous microstructure of paper as is evident from the abundant voids throughout the scaffold which can be seen in field emission scanning electron micrographs (FE-SEM) of paper substrates before and after Ni-coating (Supplementary Fig. 2a–f). The successful growth of metallic Ni was again confirmed by X-ray diffraction (XRD) analysis (Supplementary Fig. 2g) which shows an obvious peak at 44.7° corresponding to (111) plane of face centered cubic (fcc) $Ni^0$. The broad fcc (111) peaks also suggest presence of nanoparticulate Ni which further corroborate the observations made from FE-SEM analysis. The crystallite size is estimated to be ~23 nm from the Scherrer equation which is similar to the size estimated by FE-SEM analysis. Moreover, XRD indicates the presence of α, β-Ni(OH)$_2$ as indexed in Supplementary Fig. 2 which is attributed to partial surface oxidation. The formation of surface hydroxide layer is also validated using X-ray photoelectron spectroscopy (XPS) analysis (Supplementary Fig. 2h–j) which reveals presence of a doublet for Ni corresponding to the Ni$2p_{3/2}$ (857.66 eV) and Ni$2p_{1/2}$ (875.4 eV) states, confirming the predominance of $Ni^{2+}$ valence form. The presence of surface hydroxyl groups is also evident from the O 1$s$ spectrum showing a peak at 531.23 eV that is commonly attributed to oxygen of the –OH groups[29]. The surface hydroxides play a dual role of acting as catalytic sites for water oxidation and as barrier coatings to prevent complete oxidation of metallic Ni within the volume of the porous conducting paper. It was indeed observed that the presence of surface hydroxyl groups hardly affects the bulk electronic conductivity of Ni-P substrate as electroless plating decreases the sheet resistance of the paper to merely 4.2 Ω sq$^{-1}$ from an initial value of $10^{15}$ Ω sq$^{-1}$ for bare cellulose paper[30]. To further investigate the role of Pd and Sn traces, individual Pd- and Sn-activated papers were fabricated and their current–voltage responses show that Pd or Sn do not contribute to the bulk electronic conductivity of Ni-P (Supplementary Fig. 1b). We would like to stress that the high conductivity of Ni-P is retained even after storing it under ambient conditions for almost 1 year. Interestingly, Ni-P is more conducting than porous graphene (12.2 Ω sq$^{-1}$)[7], ITO and FTO glass (7–14 Ω sq$^{-1}$), Au-sputtered substrates (~5 Ω sq$^{-1}$) and

comparable to CNT coated cotton (1–4 Ω sq$^{-1}$)[31,32]. When the Ni-P substrate is either fully bent for 20–100 successive cycles or twisted end-to-end, no apparent deviation from ohmic linearity of the current–voltage (I–V) plot (Supplementary Fig. 3a, b) is observed signifying high conductivity under highly-strained conditions. The bending stiffness (flexural rigidity) of a $2 \times 6$ cm$^2$ piece of Ni-P was estimated to be 4.4 g cm[33]. Ni-P also demonstrates exceptional conductance stability at a constant voltage of 0.5 V under mechanical deformation from an initial normal (180°) to the fully bent state (0°) (Supplementary Fig. 3c).

**Modified Ni-P electrodes**. To demonstrate the application of conducting Ni-P as catalytic current collectors for overall electrochemical water splitting, they were coated with the highly active non-noble catalysts of $Ni_xFe_{1-x}OOH$ for OER (abbreviated as NiFe) and $Ni_4Mo$ bimetallic alloy for HER (abbreviated as NiMo). It should be noted that these compositions are chosen as model catalysts to demonstrate proof-of-concept application of paper-electrodes owing to the following considerations. To begin with, NiFe and NiMo are known to be the best performing electrocatalysts under alkaline conditions for OER and HER, respectively. Secondly, they are inexpensive non-noble transition metal based systems, which can strongly bind to the underlying current collector due to high compatibility with the $Ni^0$ NP coated paper. These catalysts can also be easily grown on the paper electrode by using simple and scalable method of electrochemical deposition. We would like to mention that the motivation for choosing these nickel-based catalysts comes from the prior report by Jaramillo's group on extensive screening and objective comparison of several electrocatalysts with regard to their activities and stabilities under similar conditions[34].

NiFe was electrodeposited onto Ni-P from a 1:1 aqueous solution of the corresponding metal nitrates following a literature report (see Experimental section); wherein electro-reduction of $NO_3^-$ ions increases the local pH at electrode surface and the $Ni^{2+}$ and $Fe^{3+}$ ions react with the $OH^-$ ions, generated locally, to form the bimetallic oxyhydroxide layer, $Ni_xFe_{1-x}OOH$ (Fig. 1). The electrodeposited light green colored $Ni_xFe_{1-x}OOH$ film is tightly bound to Ni-P (abbreviated as NiFe/Ni-P) and could not be dislodged even after prolonged sonication. FE-SEM images of NiFe coated Ni-P shows growth of 8–10 nm thick vertically aligned nanoflakes with interconnected, macroporous topography that does not block the underlying macroporous architecture of the paper electrode (Fig. 2a, b). This interesting morphology is beneficial for electrocatalysis as it provides an abundance of exposed catalytically active sites as well as enables rapid transport of electrons along the vertically oriented nanoflakes. Energy dispersive X-ray (EDX) spectroscopy was used to further characterize the elemental composition and distribution on the NiFe/Ni-P electrode surface (Fig. 2c). Ni, Fe, and O were detected and found to distribute uniformly over the entire area, confirming homogenous deposition of a uniform NiFe catalyst layer over the Ni-P substrate. TEM analysis (Fig. 2d) shows a clear rippled nanosheet like morphology in accordance with the FE-SEM images and a semi-transparent nature on the edges, indicate growth of ultrathin nanosheets. High resolution TEM images (HR-TEM; Fig. 2d inset) also suggest completely amorphous nature of NiFe as no lattice fringes typical for $Ni^0$, $Fe^0$, or $NiFe_2O_4$ are detected. This is also supported by the selected-area electron diffraction (SAED) pattern (Fig. 2d inset) which shows broad and diffuse halo-rings with no clear diffraction spots. The XRD pattern of NiFe/Ni-P electrode (Fig. 3a) further confirms the amorphous nature of NiFe catalyst layer as no new peaks are observed besides those from the Ni-P substrate, which corroborate with the previous studies which report on the growth of

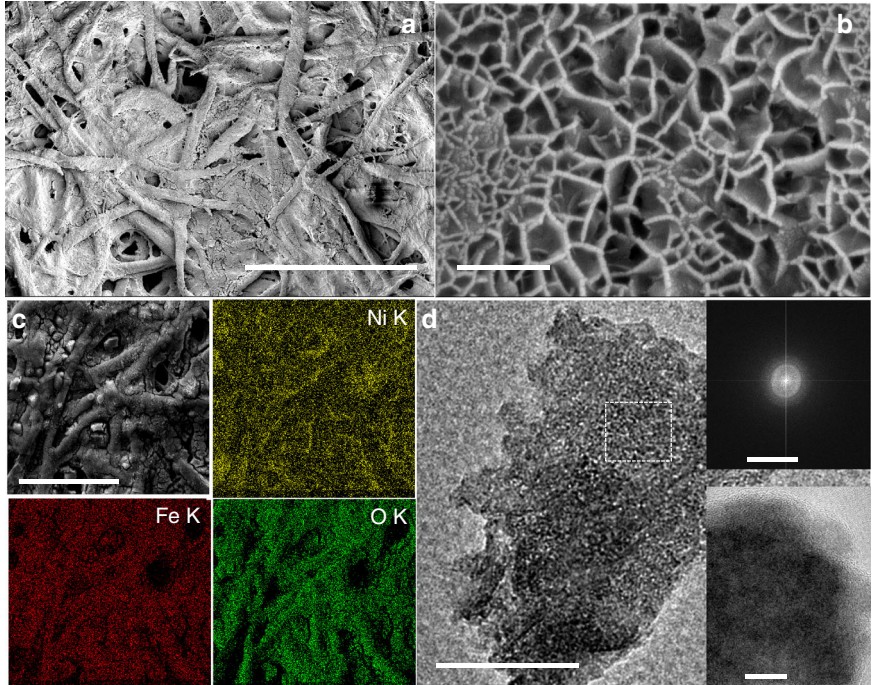

**Fig. 2** High resolution FE-SEM and HR-TEM micrographs of NiFe/Ni-P electrode. **a** Low (scale bar 100 μm) and **b** high resolution FE-SEM images of $Ni_xFe_{1-x}OOH$ film on Ni-P (NiFe/Ni-P) (scale bar 200 nm). **c** EDX elemental mapping of the NiFe/Ni-P electrode surface (scale bar 200 μm). **d** TEM micrograph showing a rippled nanosheet like morphology (scale bar 100 nm). Insets show the SAED pattern obtained from the area marked in white (scale bar 10 nm$^{-1}$) and corresponding high resolution TEM image (scale bar 10 nm), both of which confirm the amorphous nature of NiFe

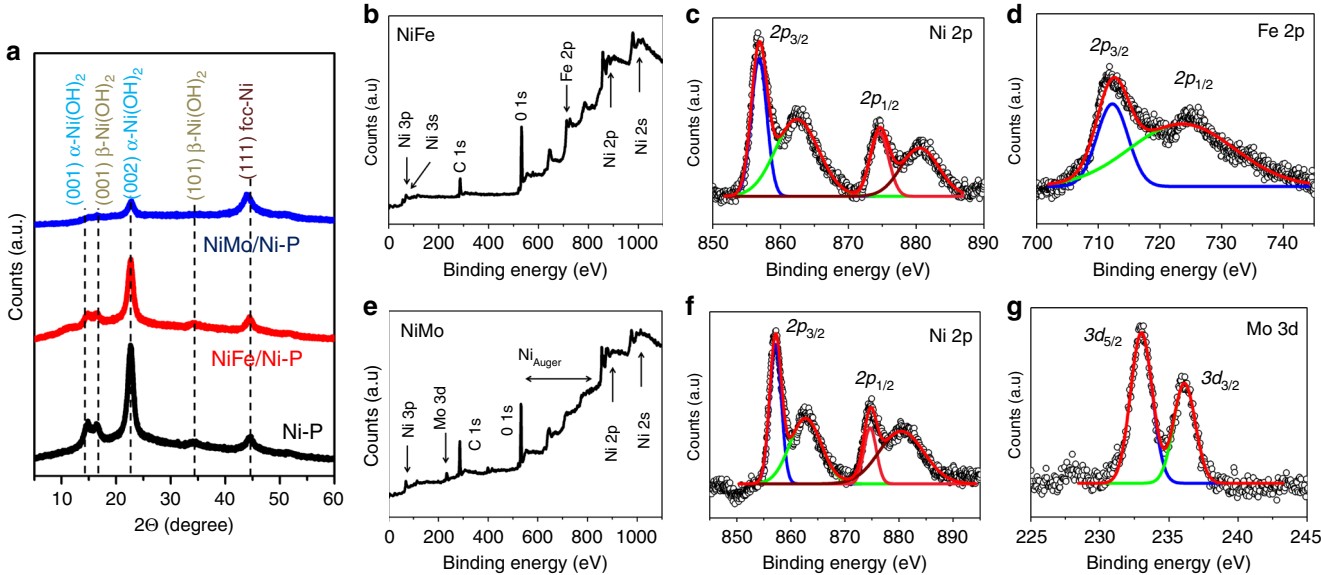

**Fig. 3** X-ray diffractograms and XPS spectra of NiFe/Ni-P and NiMo/Ni-P electrodes. **a** XRD patterns of Ni-P, NiFe/Ni-P, and NiMo/Ni-P catalysts. **b** XPS survey scan of NiFe/Ni-P with the deconvoluted core level spectra of **c** Ni 2p and **d** Fe 2p. **e** XPS survey scan of NiMo/Ni-P with the deconvoluted core level spectra of **f** Ni 2p and **g** Mo 3d

amorphous NiFe via chronoamperometry[35]. In fact, it has been proposed that amorphous NiFe electrocatalysts are much more active than their crystalline counterparts due to their structural flexibility and a high density of co-ordinatively unsaturated sites that help in the adsorption of oxidized intermediates[29]. The chemical compositions of NiFe paper-anode was further characterized by XPS analysis. The XPS spectra show the presence of C, Ni, Fe and O on the NiFe/Ni-P electrode surface (Fig. 3b). The Ni 2p spectrum could be fitted into two separate

peaks at 856.2 and 874.7 eV corresponding to the spin-orbit states of Ni $2p_{3/2}$ and Ni $2p_{1/2}$, respectively (with corresponding shake-up satellite peaks at 862.9 and 881.0 eV), as shown in Fig. 3c, which clearly establishes the dominant oxidation state of Ni as $Ni^{2+}$. Similarly, the high resolution Fe 2p spectrum (Fig. 3d) shows two peaks at 712.2 and 725.6 eV (satellite shake-up peak at 720.1 eV) corresponding to Fe $2p_{3/2}$ and Fe $2p_{1/2}$ states. This also confirms that Fe predominantly exists in the $Fe^{3+}$ state. The atomic ratio between Ni and Fe is found to be 3.3 by integrating

the respective XPS peaks which is very close to the Ni/Fe ratio of 3 reported previously[35]. Thus a composition of $Ni_3Fe(OOH)_x$ can be assigned to the NiFe catalyst layer.

The paper-cathode for HER was prepared by growing a layer of bimetallic NiMo alloy which is well known for its superior HER activity under alkaline conditions[34]. The catalyst layer was deposited onto Ni-P by cathodic current deposition from an aqueous solution of $Ni^{2+}$ and $Mo^{6+}$ ions. As observed from the corresponding FE-SEM images (Fig. 4a, b) the surface of Ni-P becomes more irregular after NiMo deposition. In contrast to NiFe, the electrodeposited NiMo has no distinctive morphology and is composed of large spherical aggregates which are uniformly spread throughout the Ni-P surface. Nonetheless, high magnification images do reveal the hierarchical nature of NiMo layer as the spherical aggregates are found to be in-turn composed of smaller, well-interconnected NPs (Fig. 4b). More-over, similar to NiFe/Ni-P paper-electrodes, the NiMo/Ni-P also retains the open, porous network of underlying cellulose paper since the NiMo particulates nucleate only along the cellulose fibers. This common feature of intact porosity observed for both NiFe and NiMo modified Ni-P electrodes is attributed to the presence of $Ni^0$ NPs which act as seeds for further nucleation and growth of catalyst layers deposited on the electrode surface. The elemental color mapping distribution obtained from EDX spectroscopy (Fig. 4c) shows a very steady distribution of Ni (green) and Mo (blue) species over the entire tested region which further confirms successful electrodeposition of a homogenous catalyst layer on the Ni-P substrate. TEM analysis of NiMo shows large aggregates while the TEM image clearly shows the nanoparticulate composition of the catalyst (Fig. 4d). The spherical NiMo NPs with an average size of 8–10 nm are found to be highly crystalline, displaying obvious lattice fringes with $d$-spacing 2.1 Å, consistent with the interplanar separation of (220) planes of bimetallic $Ni_4Mo$ structure[36,37]. Inset of Fig. 4d also shows the corresponding SAED pattern which confirms the crystalline nature of the NiMo alloy. The bimetallic alloy nature of NiMo catalyst was further confirmed by XRD pattern of NiMo/

Ni-P electrode (Fig. 3a) which shows 2 major peaks at $2\theta$ values of 43.8° and 22.6°. The latter is indexed to the $\alpha$-Ni(OH)$_2$ phase which is formed due to surface oxidation as observed for NiFe/Ni-P and bare Ni-P electrodes. It was noted that the former peak at 43.8° is slightly shifted to lower angles from the fcc-(111) peak of metallic Ni° which appears at 44.7° for bare Ni-P electrodes. This peak shift was attributed to the doping of larger sized Mo atoms (atomic radius 209 pm) within the fcc lattice of Ni (atomic radius 200 pm) and the consequent widening of the interplanar spacing which thereby confirms the formation of a doped NiMo bimetallic alloy (Mo doped in Ni lattice). Interestingly, XPS characterization of NiMo/Ni-P electrode (Fig. 3e–g) suggests the presence of $Ni^{2+}$ state corresponding to the appearance of two peaks at 856.1 and 874.6 eV, assigned to Ni $2p_{3/2}$ and Ni $2p_{1/2}$ states of the Ni $2p$ spectrum, respectively. Likewise, the appearance of 2 distinct peaks at 231.9 ($3d_{5/2}$) and 235.0 ($3d_{3/2}$) eV for high resolution Mo $3d$ spectrum also confirms the presence of Mo in high oxidation state as $Mo^{6+}$. The higher oxidation states of Ni and Mo atoms in the NiMo catalyst layer might appear counter-intuitive, but can be ascribed to the air oxidation of the surface NiMo layer and is consistent with previous reports on electrodeposited NiMo films[36,37]. This is also supported by the presence of $\alpha$-Ni(OH)$_2$ that is observed in the XRD pattern of NiMo/Ni-P electrode as expected (Fig. 3a). However, it is believed that this surface oxidized layer is eventually converted into the catalytically active bimetallic alloy under reductive potentials of HER. The relative atomic ratio of Ni:Mo in the electrodeposited catalyst was calculated through XPS peak integration and the ratio is found to be 4.1:1 which is in perfect agreement with the $Ni_4Mo$ bimetallic structure estimated from the lattice $d$-spacing value obtained by HR-TEM analysis as discussed above. Moreover, prior literature reports have estab-lished similar atomic ratio of Ni:Mo::4:1 for cathodically electrodeposited NiMo films[36,37]. Thus, based on the above structural characterization the NiMo catalyst layer can be assigned a composition of $Ni_4Mo$. It should be highlighted that understanding the nature of interactions between Ni and Mo

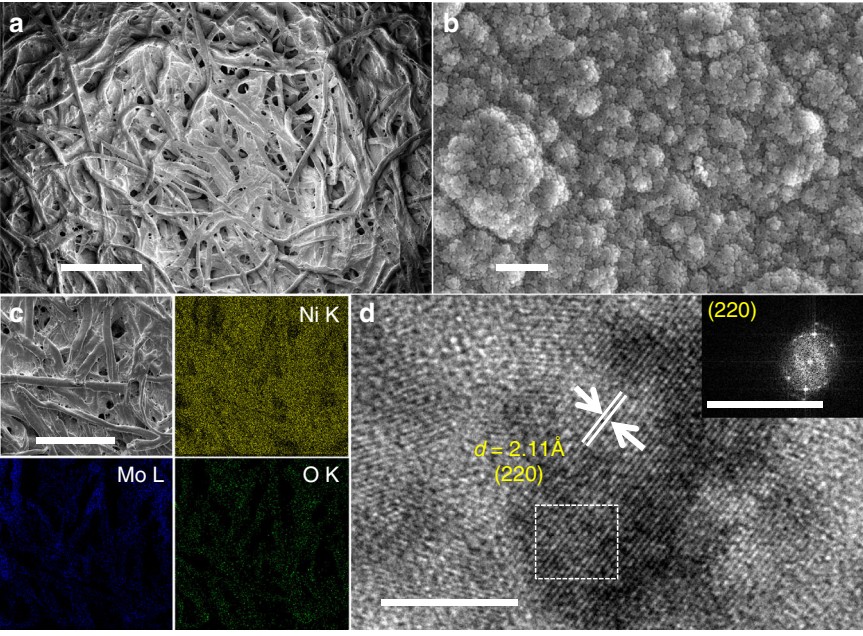

**Fig. 4** High resolution FE-SEM and HR-TEM micrographs of NiMo/Ni-P electrode. **a** Low (scale bar 100 μm) and **b** high resolution FE-SEM images of $Ni_4Mo$ nanoparticulates on Ni-P (NiMo/Ni-P) (scale bar 200 nm). **c** EDX elemental mapping of the NiMo/Ni-P electrode surface (scale bar 200 μm). **d** TEM image showing the lattice fringes of $Ni_4Mo$ phase (scale bar 5 nm) and inset shows the SAED pattern obtained from the area marked in white (scale bar 20 nm$^{-1}$)

atoms in bimetallic $Ni_4Mo$ alloy or identifying the catalytically active metal sites in the amorphous $Ni_xFe_{1-x}OOH$ lies outside the scope of this work. On the contrary, these systems are chosen as model, representative electrocatalysts to demonstrate the utility of the paper-based flexible current collector.

**Oxygen evolution reaction.** The electrochemical activities of different paper-based anodes towards OER were studied with a standard 3-electrode cell by linear sweep voltammetry (LSV) with 1 M KOH and the potentials are reported against reversible hydrogen electrode (RHE). The catalytic performance of the paper-anode was also compared to the benchmark precious metal-based $IrO_2$. The NiFe/Ni-P paper-anode shows an obvious anodic peak around ~1.35 V while $IrO_2$ starts to evolve $O_2$ at around 1.5 V in agreement with previous reports (Fig. 5a and Supplementary Table 1)[9]. In spite of the significant anodic current density at low potentials for NiFe/Ni-P, $O_2$ gas bubbles are not observed until 1.45–1.47 V, which suggests that the anodic shoulder in the low potential range 1.3–1.4 V arises from the transformation of Ni(II) to Ni(III)[38,39]. The parameter of merit commonly used for comparing different electrocatalysts is the overpotential required to achieve a current density of 10 mA cm$^{-2}$ (corresponding to a solar water splitting device with 10% efficiency)[34]. However, as is commonly observed for various Ni-based OER catalysts, the deconvolution of Ni(II)/Ni(III) process from OER is difficult because of the overlap of these two reactions within the potential window of 1.3–1.4 V v/s RHE. Therefore to unambiguously rate different OER catalysts in this work, a modified metric of overpotential required to reach 50 mA cm$^{-2}$ ($\eta^{50}$) is chosen instead of the conventional 10 mA cm$^{-2}$[40]. The $\eta^{50}$ of NiFe/Ni-P is exceptionally low at merely 241 mV (Fig. 5b) as compared to 352 mV for bare Ni-P and it reaches 100 mA cm$^{-2}$ at only 1.47 V with vigorous $O_2$ gas evolution (Supplementary Movie 1). The reaction turnover frequency (ToF) for NiFe/Ni-P at overpotential of 250 mV is estimated to be 0.13 s$^{-1}$ which is one order of magnitude higher than that of bare Ni-P electrodes (0.015 s$^{-1}$). NiFe/Ni-P paper-anode outperforms even the precious $IrO_2$ ($\eta^{50} = 434$ mV, $\eta^{10} = 320$ mV) benchmark and

registers $\eta^{50}$ value lower than the $\eta^{10}$ of many state-of-the-art OER catalysts as elaborated in Supplementary Table 2[41–53]. In order to understand and elucidate the unique role of Ni NPs on the paper surface, the OER activities of unmodified Ni-P and commercial Ni-foam are compared and contrasted. It is evident that Ni-P outperforms commercial Ni-foam as the former requires 352 mV as compared to 440 mV required for the latter to reach 50 mA cm$^{-2}$ under identical conditions. This superior performance of Ni-P is ascribed to the presence of nanoparticulate Ni$^0$ with exposed reactive surfaces as compared to bulk Ni of Ni-foam. Such high activity of simple Ni-P papers is also evidenced in case of full water electrolysis in a two-electrode device (discussed below). These observations underscore the importance of nanostructured porous surfaces for improved electrocatalytic activity (see Supplementary Discussion 1 and Supplementary Fig. 4). Interestingly, the unmodified Ni-P ($\eta^{50} = 352$ mV) also supersedes the activity of $IrO_2$ ($\eta^{50} = 434$ mV) which further establishes the inherent catalytic property of Ni NP coated onto cellulose paper. Thus, besides being an efficient flexible current collector and catalyst support the Ni-P conducting papers are also highly electrocatalytic in their own right which distinguishes them from other rather inert commercial counterparts such as Ni-foam, carbon cloth, and fiber paper etc.

The fast adsorption/desorption kinetics of the oxygenated species on the NiFe/Ni-P electrode surface is suggested from its lowest Tafel slope at 29.6 mV dec$^{-1}$ which is significantly lower than 54.8 mV dec$^{-1}$ recorded for benchmark $IrO_2$ and 78.37 mV dec$^{-1}$ for bare Ni-P electrode (Fig. 5c). Lower Tafel slope for Ni-P electrode (78.4 mV dec$^{-1}$) as compared to its commercial counterpart of Ni-foam (89.8 mV dec$^{-1}$) further confirms the role of Ni$^0$ NPs with abundant exposed active sites anchored on a porous cellulose scaffold in enabling rapid mass transport and facile reaction kinetics. The exchange current density determined by extrapolation on the basis of Tafel plot for NiFe/Ni-P is found to be $5.49 \times 10^{-6}$ mA cm$^{-2}$ which is same order of magnitude as that of $IrO_2$ ($28.9 \times 10^{-6}$ mA cm$^{-2}$). The apparent anomaly of lower exchange current density but higher OER activity for NiFe/Ni-P than $IrO_2$ is discussed in Supplementary Discussion 2.

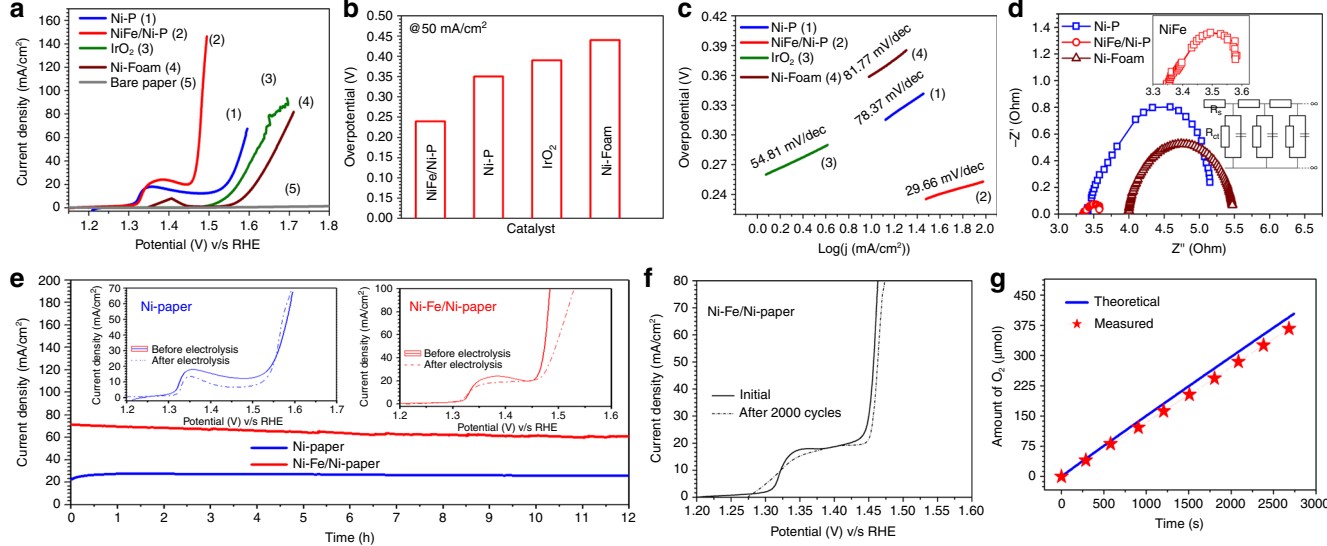

**Fig. 5** Electrochemical performance of NiFe/Ni-P electrode towards OER. **a** OER polarization curves (iR-corrected) with a scan rate of 10 mV s$^{-1}$ in 1 M KOH, **b** the potentials required to reach 50 mA cm$^{-2}$, **c** corresponding Tafel plots and **d** EIS spectra under a dc bias of 1.6 V. Insets show the Nyquist plot of NiFe/Ni-P and the equivalent transmission line model circuit used for fitting. **e** Chronoamperometric stability test of NiFe/Ni-P and Ni-P for 12 h at 1.5 V (without iR-correction). Insets show steady state LSV plots before and after 12 h. **f** LSV scan of NiFe/Ni-P electrode before and after 2000 CV cycles showing a negligible increase of $\eta^{50}$ by 0.48%. **g** Faradic efficiency measurement of NiFe/Ni-P showing the theoretically calculated and experimentally measured $O_2$ gas with time

Electrochemical impedance spectroscopy (EIS) was employed to understand the interfacial charge transfer kinetics and the corresponding Nyquist plot is fitted with a standard transmission line circuit to extract the relevant physical parameters (Fig. 5d). As expected, a dramatic drop in charge transfer resistance ($R_{ct}$) from 1.6 Ω for bare Ni-P to merely 0.3 Ω for NiFe/Ni-P is observed which clearly indicates improved charge transfer dynamics at the interface between electrode and electrolyte mediated by the NiFe active layer. Another important metric for assessing the activity of any electrocatalyst is its long term stability under operating conditions. To alleviate any concerns regarding the perceived fragile nature of the paper-electrode in a corrosive environment of 1 M KOH, rigorous stability tests were performed. The response for both NiFe/Ni-P and bare Ni-P electrodesis highly stable for up to 12 h at a bias voltage of 1.5 V with only a minor attenuation of 6.35 and 13.7% in the anodic current density for Ni-P and NiFe/Ni-P, respectively (Fig. 5e). The LSVs recorded before and after 12 h bulk electrolysis reveal almost no change in $\eta^{50}$ value for Ni-P while a small anodic shift of 20 mV is registered for NiFe/Ni-P (inset Fig. 5e). In chronopotentiometry mode at 30 mA cm$^{-2}$ (Supplementary Fig. 5a), both these electrodes sustain the applied current density without any change in their overpotentials. Two thousand cycles of continuous cyclic voltammogram (CV) scans at a rate of 100 mV s$^{-1}$ further establishes the prominent ability of NiFe/Ni-P to withstand accelerated electrochemical processes (Fig. 5f). LSV of the same electrode before and after 2000 CV cycles shows a negligible increase of $\eta^{50}$ by 0.48%. FE-SEM imaging of NiFe/Ni-P electrode after prolonged electrolysis for 12 h reveals a change in morphology of the NiFe top layer from the initial vertically aligned nanosheets to a rice-grain like structure, whereas no such morphological changes were observed for the Ni-P electrode (Supplementary Fig. 5b–g). Intriguingly, this alteration in the morphology of NiFe top layer has a minimal effect on the overall electrochemical activity of NiFe/Ni-P electrode (Supplementary Discussion 3 and Supplementary Fig. 6). It is interesting to note the robustness and durability of both the paper-electrodes even in the presence of vigorous gas bubbling form their surfaces. The

Faradic efficiency is quantified by measuring the $O_2$ gas generated during 1 h bulk electrolysis by eudiometry as described in the experimental section. By comparing the theoretical product gas calculated from the amount of charge passed at different time intervals with the experimentally measured quantities (Fig. 5g), a Faradic efficiency of 97.3% is obtained. This near unity value also suggests the consumption of almost all the charge without any associated parasitic reactions in the presence of NiFe top layer in NiFe/Ni-P electrodes. This is in contrast with Ni-P where a competing kinetically more facile 2e$^-$ oxidation of Ni$^0$ into Ni$^{2+}$ occurs apart from the sluggish 4e$^-$ water oxidation reaction (Supplementary Discussion 4).

**Hydrogen evolution reaction**. The HER performance of NiMo/Ni-P is measured in 1 M KOH in a three electrode electro-chemical cell and is compared with unmodified Ni-P, commercial Ni-foam and commercial benchmark of Pt/C (40 wt%) (Fig. 6a and Supplementary Table 3). In comparison to bare Ni-P, which shows a $\eta^{10}$ value of 128 mV, the NiMo/Ni-P requires only 32 mV to reach 10 mA cm$^{-2}$. Interestingly this value for NiMo/Ni-P electrode is very close to the corresponding overpotential of 13.3 mV for the benchmark Pt/C electrocatalyst tested under identical conditions (Fig. 6b). Furthermore, NiMo/Ni-P shows a negligible onset potential of −17 mV and tremendously enhanced cathodic current density coupled with vigorous $H_2$ release (Supplementary Movie 2). The ToF for NiMo/Ni-P at overpotential of 130 mV is estimated to be 0.33 s$^{-1}$ which is one order of magnitude higher than that of bare Ni-P electrodes (0.024 s$^{-1}$). Comparative analysis (Supplementary Table 4) shows that $\eta^{10}$ for NiMo/Ni-P is much lower than other reported non-noble HER catalysts[9,10,29,54–71]. The significant differences in activities of unmodified Ni-P and commercial Ni-foam are also worth noting. As seen in the case of OER, Ni-P performs much better than Ni-foam with a difference of more than 200 mV in their respective $\eta^{10}$ values as well as much smaller onset potentials. This improved activity of Ni-P over its commercial counterpart may be due to the nanoparticulate metallic Ni in contrast to bulk Ni of

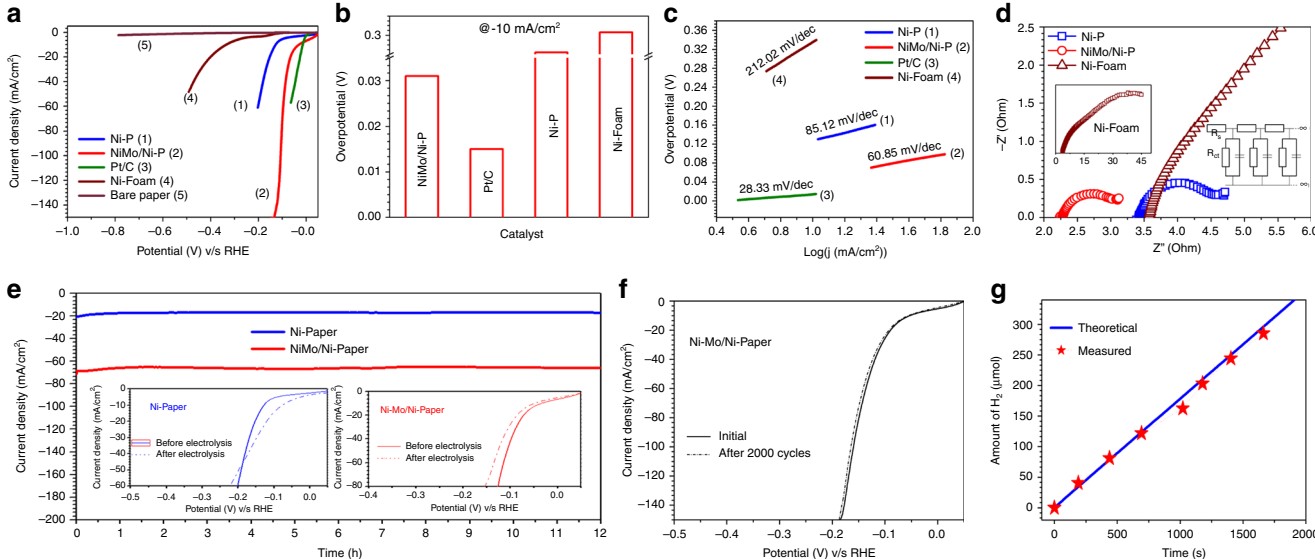

**Fig. 6** Electrochemical performance of NiMo/Ni-P electrode towards HER. **a** HER polarization curves (iR-corrected) with a scan rate of 10 mV s$^{-1}$ in 1 M KOH, **b** the potentials required to reach −10 mA cm$^{-2}$, **c** corresponding Tafel plots and **d** EIS spectra under a dc bias of −0.1 V for all catalysts. Insets show the Nyquist plot of Ni-foam and equivalent transmission line model circuit. **e** Chronoamperometric stability test of NiMo/Ni-P and Ni-P for 12 h at −0.15 V (without iR-correction). Insets show steady state LSV plots before and after 12 h. **f** LSV scan of NiMo/Ni-P electrode before and after 2000 CV cycles showing no apparent changes in the HER activity. **g** Faradic efficiency measurement of NiMo/Ni-P showing the theoretically calculated and experimentally measured $H_2$ gas with time

Ni-foam and further underlines the inherent catalytic nature of Ni-P based flexible current collectors. In fact, the promising OER and HER activity of unmodified Ni-P makes it a highly suitable bi-functional catalyst substrate for overall water splitting as is demonstrated below. Furthermore, the Pd and Sn traces in Ni-P do not contribute to the electrochemical activities and remain as mere spectator species (Supplementary Fig. 1c). Tafel analysis was performed to obtain a mechanistic understanding of HER on paper-electrodes. For NiMo/Ni-P, the Tafel slope is found to be 60.8 mV dec$^{-1}$ which is close to Pt/C (53.38 mV dec$^{-1}$) and is smaller than the bare Ni-P (85.12 mV dec$^{-1}$) (Fig. 6c). The much lower Tafel slope for unmodified Ni-P papers as compared to commercial Ni-foam establishes the rapid reaction kinetics on the former's surface and is in complete agreement with the superior performance observed for Ni-P. The Tafel slope of ~60 mV dec$^{-1}$ observed for NiMo/Ni-P electrodes suggests that HER proceeds via the formation of adsorbed intermediates according to the Eq. 3:

$$M + H_2O + e- \rightarrow M - H_{ads} + OH^- \qquad (3)$$

whereas the reaction rate is determined by the electro-desorption step according to the Eq. 4[72]:

$$M - H_{ads} + H_2O + e- \rightarrow M + H_2 + OH^- \qquad (4)$$

The fast reaction kinetics of NiMo/Ni-P results from a culmination of factors such as high intrinsic activity of NiMo, porous network of Ni-P support and rapid mass transport facilitated by quick release of gas bubbles. The exchange current density of NiMo/Ni-P (2.04 mA cm$^{-2}$) is comparable to Pt/C (2.81 mA cm$^{-2}$) and greater than bare Ni-P (0.30 mA cm$^{-2}$) (Supplementary Discussion 2). EIS studies indicate that NiMo/Ni-P electrode shows least resistance to charge transfer with $R_{ct}$ value of 0.75 Ω which is smaller than bare Ni-P electrode (1.3 Ω) and thereby attests to the excellent intrinsic activity of NiMo layer towards HER (Fig. 6d). It should be highlighted that the nanostructured Ni-P electrode shows improved charge transfer dynamics than the commercial Ni-foam as evidenced from its smaller $R_{ct}$ values. The long term durability of the paper electrodes for HER is confirmed by chronoamperometry, where Ni-P and NiMo/Ni-P electrodes were held at a constant potential of −0.15 V (without iR correction) and both show a flat current response with no apparent activity loss for at least 12 h (Fig. 6e). The voltammogram recorded after prolonged electrolysis (inset Fig. 6e) for Ni-P shows a slight increase in catalytic activity with a drop in $\eta^{10}$ to 90 mV from the initial ~130 mV, which is attributed to the reduction of surface hydroxides during the initial period of hydrogen evolution. On the contrary, the catalytic activity of NiMo/Ni-P drops marginally at higher overpotentials, resulting from an increase in mass transport resistance due to vigorous gas bubbling at higher applied bias. The steady state HER stability of Ni-P and NiMo/Ni-P was further validated by passing a constant current density of −30 mA cm$^{-2}$ which shows a flat potential response (Supplementary Fig. 7a). Furthermore, accelerated degradation studies over 2000 CV cycles registered no apparent changes in the HER activity of NiMo/Ni-P electrode with regards to the $\eta^{50}$ value (Fig. 6f) which conclusively establishes the robustness and durability of the paper-electrode. This was again supported by the intact morphology of Ni-P and NiMo/Ni-P papers even after prolonged hydrogen evolution as seen in the FESEM images of the respective electrodes after 12 h bulk electrolysis (Supplementary Fig. 7b–g). Comparison of the evolved H$_2$ gas and the theoretical value using Faraday's equation at different time intervals (Fig. 6g) yields an outstanding average

Faradic efficiency of 99.4% for NiMo/Ni-P indicating almost complete conversion of electrical energy into chemical energy.

**Intrinsic activity and flexibility**. Although the OER and HER activities were determined based on the projected area of the electrodes, the actual available surface area is larger than its geometrical area. The real to geometrical surface area ratio can be estimated from the roughness factor, σ, calculated as in Eq. 5

$$\sigma = \frac{C_{dl}}{40 \ \mu F \ cm^{-2}}, \qquad (5)$$

where $C_{dl}$ is double-layer capacitance calculated from the rate-dependent CV plots in the non-faradic region and the specific capacitance of 40 μF cm$^{-2}$ is the average double-layer capacitance for a smooth catalyst surface[40]. The intrinsic activity of different catalysts is compared by normalizing the current density at a particular overpotential to the electrochemically available surface area. From the straight-line fits of maximum current density against varying scan rates from 10 to 100 mV s$^{-1}$ extracted from CVs recorded in the non-faradic region (Fig. 7) for NiFe/Ni-P electrode, $C_{dl}$ is found to be 62.76 mF cm$^{-2}$ which is almost twice that for bare Ni-P (30.78 mF cm$^{-2}$) and is believed to originate from the additional electroactive surface created by the porous NiFe layer. However, deposition of large aggregates of NiMo layer causes a drop in $C_{dl}$ (3.6 mF cm$^{-2}$). Even after normalization with the electrochemically active surface area, both NiFe/Ni-P and NiMo/Ni-P show higher current outputs than bare Ni-P for OER and HER (Supplementary Table 5) which clearly establishes the higher intrinsic activities of NiFe and NiMo catalysts on the paper surface.

Besides high intrinsic activity, the open macroporous framework of the paper substrate facilitates instant diffusion of electrolyte species to the catalyst active sites which leads to their exceptional performance. This proposition is indeed supported by the complete overlap of polarization curves recorded at scan rates between 2 and 100mVs$^{-1}$ under otherwise identical conditions for both NiFe/Ni-P and NiMo/Ni-P (Fig. 8a). The hierarchical porosity of the composite electrode emerging from the amalgamation of macro-porous catalyst top-layer and the porous microstructure of Ni-P current collector helps in providing loose texture and open spaces that boost the fast release of evolved gas bubbles. This excellent mass transport enabled rapid reaction kinetics is also predicted by the Tafel analysis as discussed before. To demonstrate the flexibility of NiFe/Ni-P and NiMo/Ni-P paper electrodes their activity was also tested under various conditions of mechanical deformation. A bending stress was applied to the electrode by controllably varying the curvature angle between 0–180°. The LSV polarization curves completely overlapat all bending angles (Fig. 8b) and show very similar overpotentials. Importantly, the robust yet flexible electrodes present no evidence of structural failure and catalytic activity loss even after being bent ten times at every curvature angle. To understand the generalizability of our approach to other commonly available flexible substrates (Supplementary Discussion 5), the same electrodes were fabricated on cotton fabric (Supplementary Fig. 8) and these electrodes perform much more efficiently than many reported non-noble catalysts (Supplementary Fig. 1).

**Overall water splitting**. Having established the general applicability of our methodology, the paper based electrodes were tested for their overall water splitting performance. For this purpose, two-electrode alkaline electrolyzer cells were constructed with NiFe/Ni-P anode and NiMo/Ni-P cathode (NiFe-NiMo for brevity), and separately with bi-functional Ni-P electrodes (denoted as bi-Ni-P). For a two-electrode configuration charge

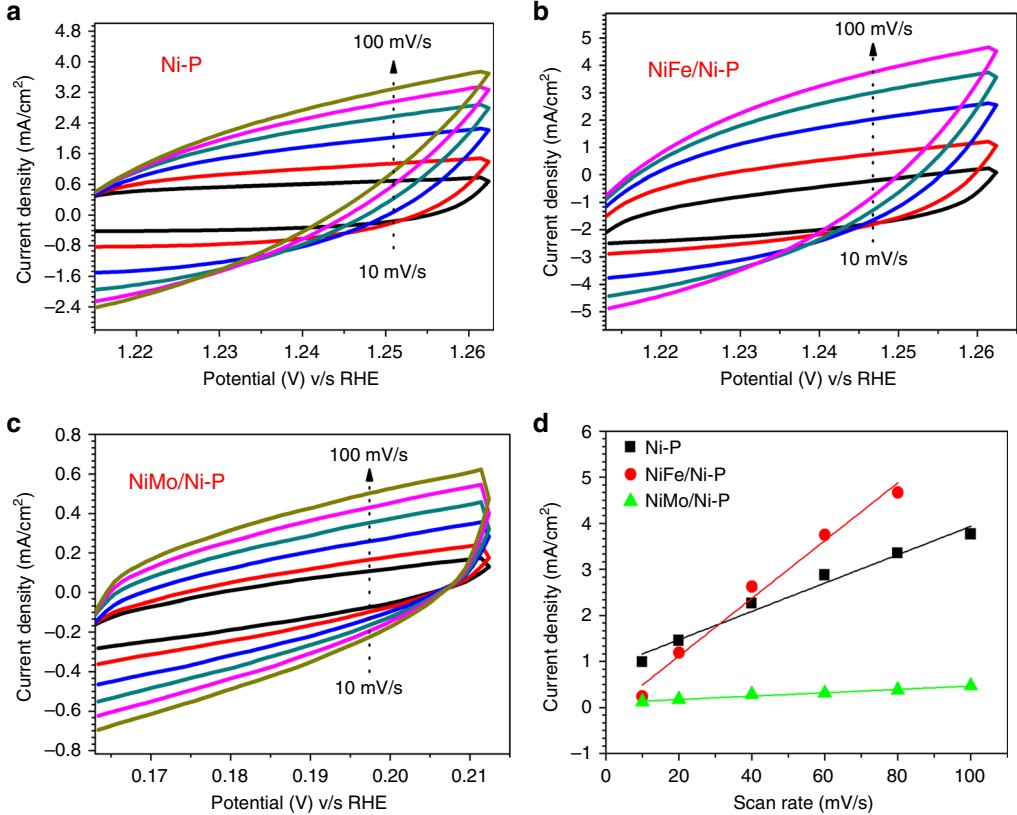

**Fig. 7** Electrochemical active surface area using CVs recorded in non-faradic region at different scan rates. **a** Ni-P; **b** NiFe/Ni-P; **c** NiMo/Ni-P; **d** Straight line fits of maximum anodic current density versus scan rate plots for Ni-P, NiFe/Ni-P and NiMo/Ni-P

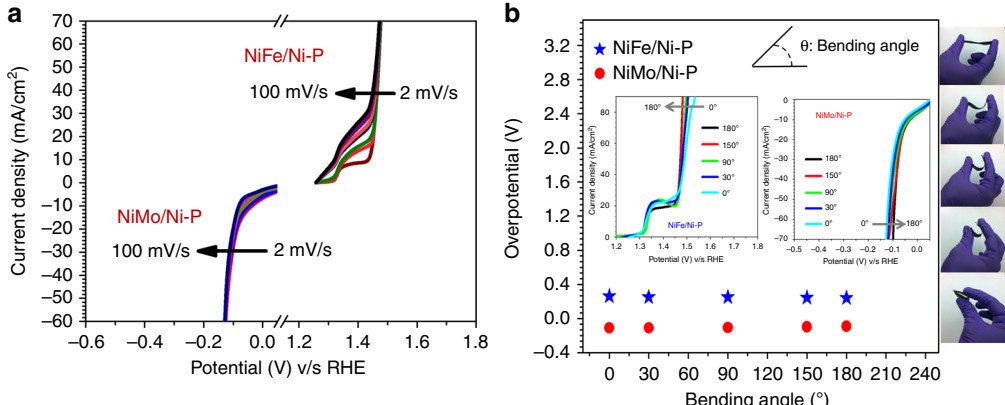

**Fig. 8** Scan rate dependence of electrode activity and their performance under bending deformations. **a** Steady state polarization curves at varying scan rates of 2, 5, 10, 15, 20, 25, 50, 75, and 100 mVs$^{-1}$ for NiFe/Ni-P and NiMo/Ni-P under OER and HER conditions, respectively. **b** Flexibility studies of NiFe/Ni-P and NiMo/Ni-P electrodes showing the overpotentials required to reach current densities of 50 and −10 mAcm$^{-2}$, respectively at different bending angles with corresponding digital images. Insets show full steady state LSVs at different bending angles along with the definition of 'bending angle' (top)

conservation requires equal anodic and cathodic current density, but with an opposite sign. Therefore, the net cell voltage required to achieve a particular current density for total water splitting is the voltage difference (ΔV) between the respective half reactions. To verify this, steady state polarization curves were recorded in 1 M KOH and it was found that the ΔV curves plotted by subtracting the voltages of individual OER and HER reactions (Fig. 9a) overlap satisfactorily with the iR-corrected LSVs of respective electrolyzers for both NiFe-NiMo and bi-Ni-P couples (Fig. 9b). The steady state LSVs (without iR correction) for both these systems show the Ni(II)/Ni(III) oxidation shoulder in the

low potential region of 1.2–1.4 V (Fig. 9c). For NiFe-NiMo, catalytic current onset is observed at ~1.45 V, however appreciable gas-bubbles start evolving only at ~1.5 V. A current density of 10 mA cm$^{-2}$ could be reached at considerably low voltage of 1.51 V representing $\eta^{10}$ of only 280 mV for coupled water oxidation and reduction. 280 mV is much lower than that for bi-Ni-P electrolyzer (460 mV), benchmark IrO$_2$-Pt/C (~370 mV)[67,68], and other non-precious electrocatalysts such as NiFe/NiCo$_2$O$_4$/NF (440 mV), porous MoO$_2$ (300 mV) and NiP@CP (400 mV) (Supplementary Table 6)[9,10,67–71,73–82]. Excitingly, to the best of our knowledge $\eta^{10}$ of 280 mV is also amongst the lowest reported

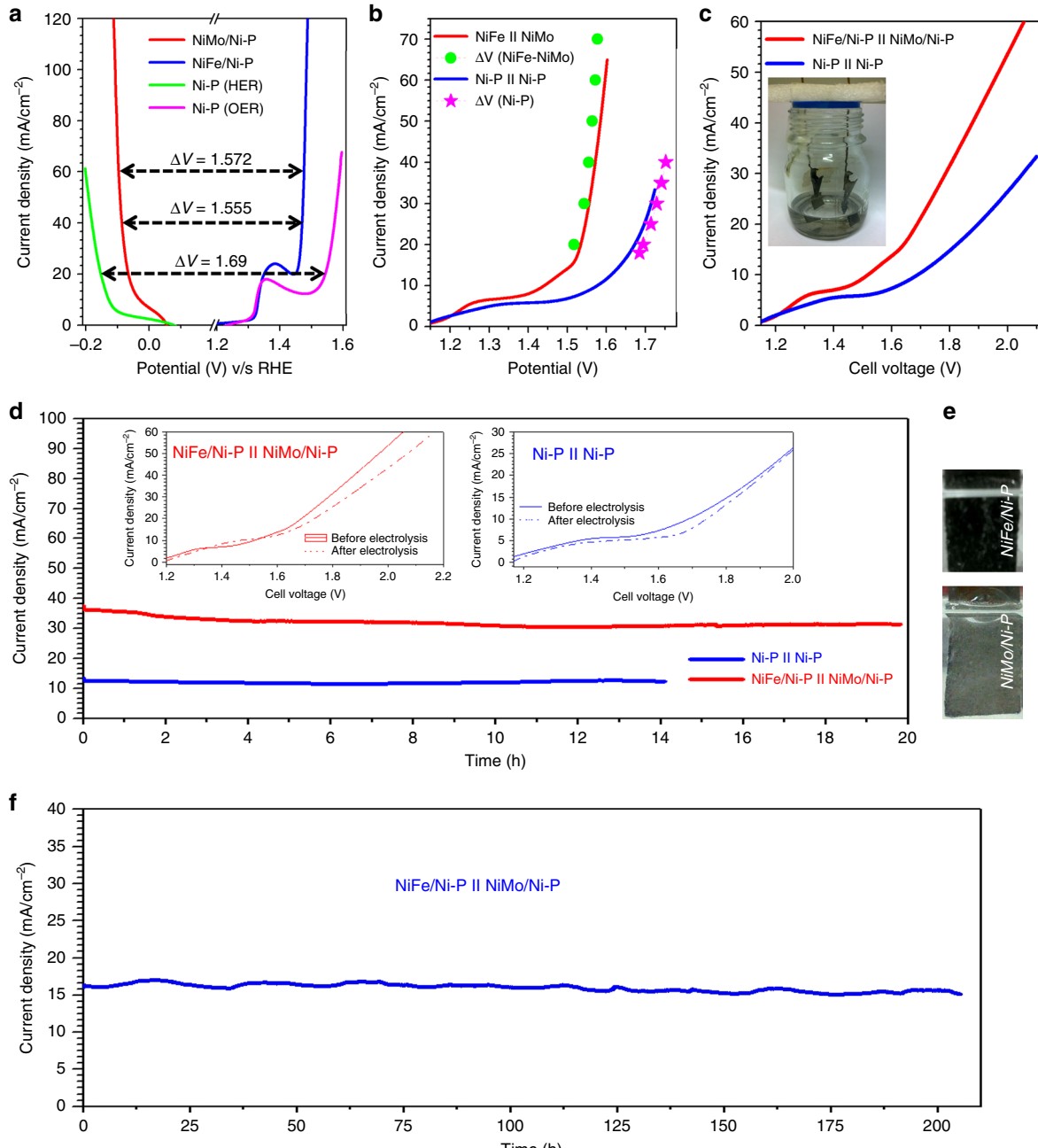

**Fig. 9** Electrocatalytic performance of 2-electrode cell and long-term stability under continuous operation. **a** Steady state polarization curves of Ni-P, NiFe/Ni-P and NiMo/Ni-P under OER and HER conditions with corresponding voltage differences at fixed current densities. **b** Steady state polarization curves of NiFe-NiMo and bi-Ni-P electrolyzers (with iR-correction) in 1 M KOH along with corresponding voltage difference plots. **c** The analogous polarization curves without iR-correction. Inset showing the assembly of an electrolyzer. **d** Chronoamperometric stability test of NiFe-NiMo and bi-Ni-P electrolyzers in 1 M KOH at 1.75 V. Inset shows steady state LSVs before and after bulk electrolysis. **e** Digital images of the electrodes under operating conditions showing release of gas bubbles. **f** Long term chronoamperometric stability test of NiFe-NiMo electrolyzer in 1 M KOH at 1.7 V in the practical device condition i.e., near 10 mA cm$^{-2}$

values for full water splitting under alkaline conditions with earth abundant flexible catalysts. The energy conversion efficiency of NiFe-NiMo, estimated at 98.01% using the 'thermoneutral potential' of water electrolysis, i.e., 1.48V[69,83], is not only higher than most of the non-noble couples but also than that of IrO$_2$/Pt measured at same current density[71]. It is also interesting to take a note of the exceptional activity of 87.5% for bare Ni-P current collectors which act as bi-functional electrocatalysts towards overall water splitting aided by their macroporous topography and nanostructured active sites, as alluded to previously.

Long term stability of NiFe-NiMo and bi-Ni-P electrolyzers was studied in a potentiostatic mode by maintaining a constant cell voltage of 1.75 V for at least 20 h. Vigorous gas bubbling was observed on both the electrodes after an initial period of ~5 min (Supplementary Movie 3). The corresponding chronoamperometric responses (Fig. 9d) show extremely stable current outputs for both the cells with a negligible attenuation of 2.11 and 13.8% recorded at the end of 20 and 12 h for NiFe-NiMo and bi-Ni-P cells, respectively. This outstanding stability even at high current densities is also evident from the almost identical line shapes of

voltammograms recorded at the end of the long-term bulk electrolysis experiment (Inset Fig. 9d). Furthermore, the robustness of both NiMo/Ni-P and NiFe/Ni-P electrodes after 20 h bulk electrolysis at relatively high current density is evident from the FE-SEM micrographs which show an intact morphology of paper electrodes with only small structural changes in the morphology of NiFe catalyst layer (Supplementary Fig. 10). Several alkaline electrolyzers were fabricated with both NiFe-NiMo and bi-Ni-P couples and similar water splitting performance was observed, illustrating high reproducibility of paper-based electrolyzers.

For a more persuasive and acceptable test of long term electrochemical water splitting, we performed bulk electrolysis in 1 M KOH using NiFe/Ni-P and NiMo/Ni-P two electrode system (Fig. 9e) and monitored the chronoamperometric trace for more than 200 h (Fig. 9f). The stable current output without any significant attenuation even for such extended time periods clearly establishes the exceptional and robust electrocatalytic activity of these electrodes which is comparable to benchmark systems in the literature[68]. Additionally; similar long term stability tests for more than 200 h were also performed for bare Ni-P electrodes (Supplementary Fig. 11) which also reveal the exceptional stability for unfunctionalized Ni-P current collectors. These exhaustive tests provide a rigorous proof for the remarkable stability and robustness of Ni-paper based electrodes, besides those already demonstrated under harsh environments such as 10 M KOH.

**Water splitting under extreme conditions**. The mechanical robustness of the paper electrode systems were finally tested under extreme conditions of 10 M KOH and high current densities, conditions which are quite often encountered in commercial alkaline electrolysis cells (Fig. 10). Figure 10a shows the OER and HER polarization curves for NiFe/Ni-P and NiMo-Ni-P, respectively. Evidently, OER starts at a lower onset potential with $\eta^{50}$ value of merely 170 mV, which is 70 mV lower than that in 1 M KOH and reaches a current density of 120 mA cm$^{-2}$ at only 1.43 V v/s RHE. The NiMo/Ni-P also retains its excellent activity towards HER under these conditions with $\eta^{10}$ value of 60 mV and reaching 150 mA cm$^{-2}$ at barely 0.17 V v/s RHE. Thus it is clear that even such highly corrosive conditions are not detrimental to the overall performance of paper-electrodes. To further rigorously test this fact, the individual electrodes were subjected to an extended durability test at considerably large current densities >10 mA cm$^{-2}$. From Fig. 10b it is obvious that both the electrodes deliver a virtually flat current response with no signs of activity loss for at least 12 h. Notably, even after continuous operation for such long periods of time under extreme conditions, the electrodes retain their structural integrity (Supplementary Fig. 12) with no noticeable mechanical failures. The electrodes were then integrated into an electrolyzer device and the combined activity of the cell was measured in 10 M KOH. From the LSV in Fig. 10c, the excellent water splitting ability of the cell is apparent. In fact a slightly improved activity is observed as $\eta^{10}$ dropped to 260 mV corresponding to an energy efficiency of staggering 99.32%. The stability of the cell was also measured in 10 M KOH at an applied potential of ~1.8 V (Fig. 10d). The electrodes display exceptional stability delivering persistent current during the entire session of 12 h which further establishes the superlative catalytic activity and mechanical robustness of the paper electrode system.

In summary, this work demonstrates a value added transformation of common substrates such as cellulose paper or cotton

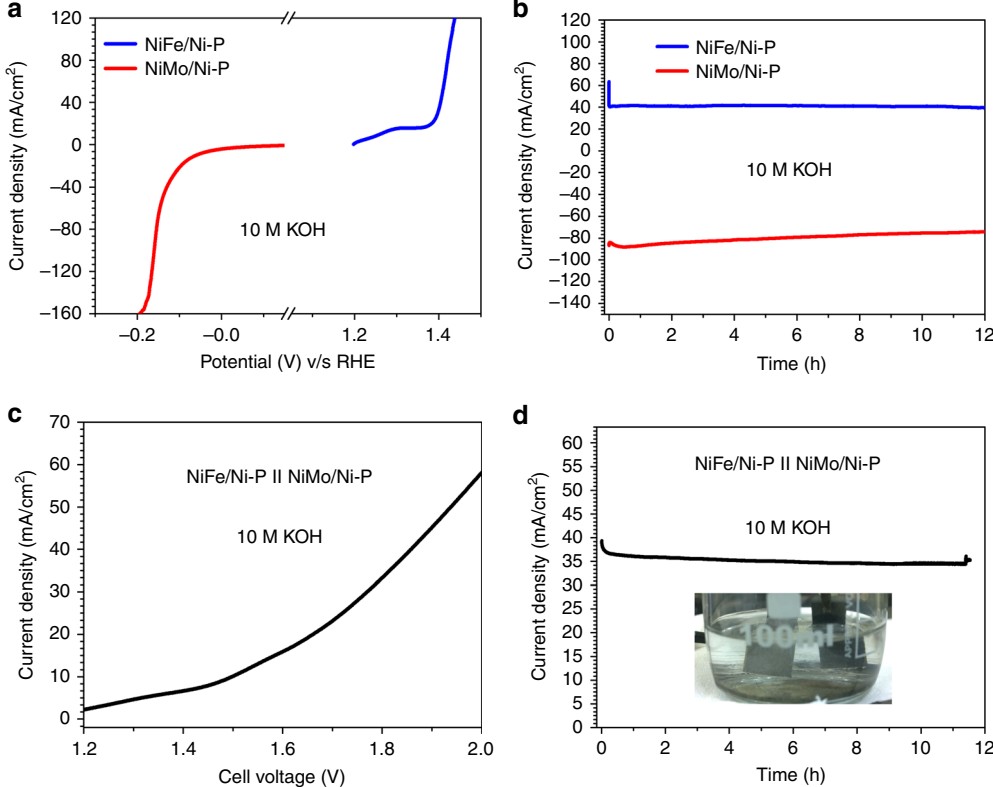

**Fig. 10** Electrocatalytic activity of paper-electrodes under harsh conditions of 10 M KOH. **a** Steady state polarization curves for NiFe/Ni-P under OER conditions and NiMo/Ni-P under HER conditions in 10 M KOH; **b** Chronoamperometric response of NiFe/Ni-P at 1.45 V versus RHE and NiMo/Ni-P at −0.15 V versus RHE for 12 h in 10 M KOH; **c** Steady state polarization curve of NiFe-NiMo electrolyzer in 10 M KOH; **d** Chronoamperometric response of NiFe-NiMo electrolyzer at 1.8 for 12 h in 10 M KOH

fabric into high efficiency flexible current collectors for overall water splitting. The use of easily available substrates, low temperature electroless plating and high scalability and reproducibility makes this method particularly attractive and cost-effective. The extraordinary OER and HER performances of composite NiFe/Ni-P and NiMo/Ni-P electrodes with favorable kinetics, robust stability in excess of 200 h and flexibility are closely related to the distinctive electrode configuration and hierarchical porosity that guarantees intimate electrical connections and improved mass and charge transport. Moreover, the unmodified Ni-papers, besides acting as flexible current collectors, have inherent bi-functional activity for both OER and HER owing to their nanostructured porous surface which distinguishes them from inert commercial counterparts such as metal foams, metal foils, carbon fibers etc. Excitingly, the NiFe-NiMo based "paper-electrolyzer" splits water with an impressive efficiency of ~98% corresponding to a cell voltage of merely 1.51 V at 10 mA cm$^{-2}$ which, to the best of our knowledge, is amongst the lowest reported values for alkaline water splitting. We believe that this work will pave a path for designing multi-functional flexible electrodes from commonly available substrates for other low-cost energy conversion and storage devices (see Supplementary Discussion 6 and Supplementary Fig. 13 for preliminary results).

## Methods

**Materials**. Tin chloride (SnCl$_2$; 98%, Sigma-Aldrich), Palladium chloride (PdCl$_2$; 99.8%, Sigma-Aldrich), nickel chloride hexahydrate (NiCl$_2$.6H$_2$O; 98%, Merck), trisodium citrate di-hydrate (Na$_3$C$_6$H$_5$O$_7$.2H$_2$O; 98%, Merck), ammonium chloride (NH$_4$Cl; 97%, Merck), sodium hypophosphite (NaH$_2$PO$_2$; 95%, Merck), aqueous ammonia solution (aq.NH$_3$; 27 wt%, Merck), nickel nitrate hexahydrate (Ni (NO$_3$)$_2$.6H$_2$O; 98%, Merck), iron nitrate nonahydrate (Fe(NO$_3$)$_3$; 98%, Merck), nickel sulfate hexahydrate (NiSO$_4$.6H$_2$O; 98%, Merck), ammonium molybdate ((NH$_4$)$_6$Mo$_7$O$_{24}$; 97%, Merck), sodium chloride (NaCl; 99%, Merck), Potassium Hydroxide pellets (KOH; 97%, Merck), hydrochloric acid (HCl; ~35%, Merck), sodium hydrogen citrate (Na$_2$C$_6$H$_6$O$_7$; 98%, Merck), potassium hexachloroiridate (K$_2$IrCl$_6$; 99%, Sigma-Aldrich), Pt/C (40 wt%, Sigma-Aldrich) and Ni-foam (Marketech International, USA) were used without further purification.

**Preparation of Ni-P and Ni-CF electrode**. A 2 × 6 cm sized piece of Whatman laboratory filter paper was first immersed into a sensitizing solution of 0.05 M SnCl$_2$ and 0.15 M HCl prepared in de-ionized (DI) water. After 30 min sensitization, the paper was rinsed copiously with DI water and acetone and dried in air. The sensitized paper was dipped in an aqueous solution of 0.6 mM PdCl$_2$ and 0.03 M HCl for 30 min and washed with DI water and acetone three times before drying in hot oven at 60 °C. Finally the Pd activated paper was immersed into an electroless plating bath held at 80 °C, consisting of 7 mmoles of NiCl$_2$, 10 mmoles of Na$_3$C$_6$H$_5$O$_7$, 30 mmoles of NH$_4$Cl, 5 mmoles of NaH$_2$PO$_2$ and the pH was adjusted with 27 wt% aq. NH$_3$ solution. After electroless plating, the paper was washed thoroughly with DI water and acetone multiple times and dried at 60 °C for 15 min. The mass loading of Ni for dried Ni-P substrate was calculated to be 3.2 mg cm$^{-2}$ which correspond to a mass loading of 32.7%. An exactly identical method was followed to prepare Ni-CF electrodes by using a cotton fabric substrate instead of paper.

**Preparation of NiFe/Ni-P and NiFe/Ni-CF electrodes**. NiFe was deposited onto a freshly prepared Ni-P or Ni-CF electrode from an aqueous electrolyte bath comprising 3 mM Ni(NO$_3$)$_2$.6H$_2$O and 3 mM Fe(NO$_3$)$_3$. 9H$_2$O following a reported method, with slight modifications[35]. The electrodeposition was performed in a potentiostatic mode with a standard 3-electrode configuration consisting of Ni-P working electrode, Pt wire counter electrode and Ag/AgCl (3 M KCl) as reference electrode. The Ni-P or Ni-CF working electrode was held at a constant potential of −1.7 V for 750 s as the electrode surface turned green from an initial ash-gray color with concomitant bubble generation. After electrodeposition, the working electrode was carefully removed and washed thoroughly with copious amounts of DI water and absolute ethanol, followed by drying in air at 60 °C for 30 min. Subsequently the final catalyst loading was determined to be 0.15 mg cm$^{-2}$ by weighing the dried electrode.

**Preparation of NiMo/Ni-P and NiMo/Ni-CF electrodes**. Bimetallic NiMo alloy was electrodeposited onto a freshly prepared Ni-P or Ni-CF substrate in a galvanostatic mode from an electrolyte bath comprising of Ni(SO$_4$)$_2$ (0.017 M), Na$_3$C$_6$H$_5$O$_7$(0.016 M), (NH$_4$)$_6$Mo$_7$O$_{24}$(0.36 mM) and NaCl (0.28 M) following a reported method with slight modifications[37]. The pH of this solution was adjusted to 9.5 with 27 wt% aq. NH$_3$ solution. The electrodeposition was performed in a

standard electrochemical cell consisting of Ni-P working electrode, Pt wire counter electrode and Ag/AgCl (3 M KCl) as reference electrode and at a constant cathodic current density of −100 mA cm$^{-2}$ for 3600 s. A dark-gray film uniformly coated the Ni-P surface with simultaneous generation of bubbles. After the deposition, the electrodes were carefully removed and washed multiple times with DI water and absolute ethanol before drying in an air oven at 60 °C for 30 min. Subsequently the final catalyst loading was determined to be 0.16 mg cm$^{-2}$ by weighing the dried electrode.

**Synthesis of IrO$_2$**. Synthesis of IrO$_2$ was adopted from the literature report[84]. K$_2$IrCl$_6$ (0.2 mmol) was added to 50 ml aqueous solution of 0.16 g of sodium hydrogen citrate (0.63 mmol). The red-brown solution was adjusted to pH 7.5 by NaOH (0.25 M) and heated to 95°C with constant stirring. After 30 min, the solution was cooled to room temperature and NaOH solution was added to adjust pH 7.5. The above steps were repeated until the pH was stabilized at 7.5. The colloidal IrO$_x$ was precipitated by centrifugation and dried overnight before calcination at 400 °C for 30 min.

**Physical characterization**. The FE-SEM images were recorded on Carl Zeiss SUPRA 55VP FESEM. The XRD measurements were performed on a PANalytical X'PERT PRO instrumenthaving Cu Kα = 1.54059 Å radiation. XPS measurements were carried out in a photoelectron spectrometer from VSW Scientific Instruments, by mounting the vacuum dried samples on copper stubs with silver paste and irradiating with Al Kα radiation (1486.6 eV). The base pressure was maintained at 5 × 10$^{-10}$ mbar. The XPS data were fitted with fityk software. XRD measurements were performed with a Rigaku (mini flex II, Japan) powder X-ray diffractometer with Cu Kα = 1.54059 Å radiation. HR-TEM images were obtained by UHR-FEG-TEM, JEOL, JEM 2100 F model using 200 kV electron source. Samples were electrodeposited on FTO substrates and samples were prepared by scratching off NiFe and NiMo from the FTO surface and drop casting the ethanol solutions onto Cu-grids. Electrical conductivity measurements were performed using standard 4-probe technique on a 0.5 × 0.5 cm$^2$ sized Ni-P or Ni-CF electrodes.

**Electrochemical characterization**. All electrochemical tests except chronopotentiometry experiments were performed in a 2-channel electrochemical workstation supplied by BioLogic Scientific Instruments, VSP-300. The various paper, cloth based electrodes (1 cm × 1.5 cm) or catalyst coated glassy carbon were used as the working electrode; a platinum wire as the counter electrode; Ag/AgCl (3 M KCl) as the reference electrode. All measurements were performed in 1 M KOH aqueous electrolyte at a scan rate of 10 mV s$^{-1}$ to minimize the capacitive current, unless stated otherwise. The catalyst inks of IrO$_2$ and Pt/C (40 wt%) powders were prepared by mixing 1 mg of the catalyst with 300 μl of absolute ethanol and 5 μl of Nafion solution and sonicated vigorously for 30 min to form a stable suspension. Four microliter of this ink was drop casted onto a GC disk (0.07 cm$^2$) and left to dry in ambient air. The final catalyst loading was 0.18mgcm$^{-2}$. Commercial Ni-foam was sonicated with soap water, DI water, 1 M H$_2$SO$_4$ and acetone respectively for 15 min each before electrochemical measurements. All potentials in the LSV polarization curves were 85% iR-corrected against the ohmic resistance of the experimental set-up using the current-interrupt method, unless stated otherwise. All the reported current densities were based on the geometrical surface area of the electrode. EIS measurements were performed with the working electrode biased at a DC potential of 1.6 V versus RHE for OER and −0.1 V versus RHE for HER while sweeping the frequency from 100 kHz to 10 mHz and maintaining an AC amplitude of 10 mV. The measured impedance data were fitted to simplified Randle's circuit to extract the series and charge transfer resistance. The electrochemical surface area and roughness factor for working electrode was measured at scan rates of 10, 20, 40, 60, 80, and 100 mV s$^{-1}$ in the non-faradic region. Flexibility tests for OER and HER were performed by bending the electrode ten times each at angles of 0, 30, 90, 150, and 180 degrees before recording the LSVs. The Faradic efficiency of different electrodes was measured by the method of eudiometry in a home-made set-up. The working electrode was fixed inside an inverted burette filled with the electrolyte. All the wires and leads connecting the working electrode were coated with an insulating varnish to prevent loss of charge through side reactions. Bulk electrolysis was performed by biasing the paper-electrodes at a fixed potential. The evolved gas is directly collected in the headspace of the inverted burette and the corresponding gas volume is determined by displacement of the vertical water column. Ideal gas approximation was used to determine the moles of gas evolved at different time intervals. The Faradic efficiency was calculated from the total charge passed through the cell at corresponding time intervals using Faraday's law, equation 6:

$$\text{Faradic efficiency} = \frac{nF \times m}{Q}, \qquad (6)$$

where $F$ = Faraday's constant (96,485.33 As mol$^{-1}$), $n$ = 2 and 4 for HER and OER respectively, $m$ = moles of gas produced and $Q$ = amount of charge passed. The reaction ToF was estimated using the method described by Yeo and Bell according

to which electrochemical equivalent ToF is given by Eq. 7[85]:

$$ToF = \frac{i \times N_A}{n \times F \times N_{atoms}}, \qquad (7)$$

where $i$ is the current, $N_{atoms}$ is the number of atoms or active sites, $n$ is the number of electrons transferred, $F$ is the Faraday's constant and $N_A$ is Avogadro's number. Here $N_{atoms}$ is estimated from the charge storage capacity of the main Ni(II)/Ni(III) redox peak. These ToF values were calculated using the assumption that all redox active Ni sites contribute to catalytic current observed in the LSV[85]. As such, the reported values are a lower bound on the ToF. The overall water splitting test was performed in a two-electrode system in 1 M and 10 M KOH and the LSVs were reported without iR correction, unless mentioned otherwise. All potentials were converted against RHE using the Nernst law, equation 8:

$$E_{RHE} = E_{Ag/AgCl} + 0.21 + 0.059 \times pH, \qquad (8)$$

where $E_{RHE}$ is the potential referred to RHE and $E_{Ag/AgCl}$ is the measured potential against Ag/AgCl (3MKCl) reference electrode.

**Data availability**. The data that support the findings of this study are available from the corresponding author upon reasonable request.

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

## Acknowledgements

A.S. thanks the academic and research fund of IISER Kolkata for his fellowship. The authors duly acknowledge the help of Mr. Manjunath Chatti with electrochemical tests of Ni-foam and for providing Pt/C sample, Mr. Sujeet Chaudhary for help with 4-probe measurements. The Department of Science and Technology (DST)—Science and Engineering Board (SERB) is duly acknowledged for the financial support under sanction no. EMR/2016/001703.

## Author contributions

A.S. fabricated the flexible electrodes, performed experiments and analyzed the data. H. D. performed I-V measurements, long term stability tests and prepared samples for electron microscopy. R.M. performed TEM imaging and fabricated the wearable paper-based Zn-air battery. A.S. and S.B. conceived the study and co-wrote the paper. All authors have given approval to its final version.

## Additional information

**Competing interests:** The authors declare no competing interests.

