## [Peer Review File · Nature Communications]

Reviewers' comments:

Reviewer #1 (Remarks to the Author):

This study reported an innovative way to transform various common substrates into flexible and efficient electrodes for water splitting. The results implied that cellulose paper and cotton have important advantages in many energy storage system regarding inexpensive, hierarchical porosity, intrinsic flexibility and tunable functionality when combined with other active materials. This work also demonstrated that high-performance water-splitting electrodes can be achieved by unique electrode configuration. Considering the importance of water splitting and the potential for practical applications, the manuscript could be published in Nature Common. after minor revisions as noted below:

- 1, The mass loading and weight ratio of Ni on the substrate are important factor for considering the conductivity and microstructure of the target electrodes, the authors should clearly point out these detailed values in the manuscript.
- 2, The Ni nanoparticles on the cotton substrate had an average size of 32 nm which is larger than that of 23 nm on the cellulose paper substrate, and the corresponding Ni-cotton electrodes showed worse water splitting performance than Ni-cellulose paper. Is there any "nanosize-effect" in the water-splitting system?
- 3, The mass loadings of catalysts should be given rather than the mass.
- 4, The authors should check the manuscript carefully since there has some formant problems in the main text, for example, Page2, lin 31; Ref 13 and 27.

Reviewer #2 (Remarks to the Author):

The authors report the synthetic method that leads to a novel current collector comprising Ni-modified paper. The material is well characterized and its further application as a support material for water splitting is also well characterized. However, I cannot recommend the publication of this paper in Nat. Commun. The reasons are listed as below:

- 1) Flexibility of the electrode is not required by water splitting devices. Thus, one of the motivations of this work is not reasonable.
- 2) The Ni-P material is not advantageous over other commercially available collector materials, such as Ni foil, Ni foam and stainless steel. The preparation of Ni-P material is complex, with the use of Pd as the catalyst.
- 3) Ni-Fe hydroxides and NiMo alloys are well established electrodes for OER and HER. So, the core content of this paper is lack of novelty.

Overall, I do not recommend the publication of this paper in Nat. Commun.

Reviewer #3 (Remarks to the Author):

The design and development of effective catalytic materials and corresponding OER/HER electrodes play profound roles in the electrocatalytic water splitting. The manuscript reported a combinatorial methodology for fabricating the flexible NiFe hydroxide/Ni-paper OER and NiMo/Ni-paper HER electrodes and demonstrated their exceptional performance for electrocatalytic water splitting in terms of their lower over-potentials and bulk stability. Through the typical characterisations, the authors claimed the excellent activities were due to the selected catalysts and the nanostructured porous surface of the electrodes which largely related to the Ni-paper substrates. The research topic well fits the scopes of the Natcomm.

The ideas of the combination of electroless plating of Ni on paper to form the Ni-Paper substrates and using the obtained substrates to electroplate the state-of-the-art catalysts are smart, though the methodology is not novel. The context is not concise and significant weakness exists in the manuscript as detailed following.

- 1.) The paper addressed the flexibility and the exceptional activity of the Ni/paper substrates for OER and HER electrodes, yet the understanding of the underpinning improvements of OER and HER reaction was not confidently attempted nor persuasive.
- 2.) In the experimental section, the important parameters, the sizes of the Ni-P substrates, were not presented for OER and HER electrodes.
- 3.) The comparisons of the overpotential based on the 10mA/sq or 50 mA/sq currents were conducted either in different electrolytes or different electrode sizes (tables in the context and supporting information), which shakes the foundation of the claims to exceptional performance of the flexible electrodes.
- 4.) Stability of the electrodes is the pivotal property for OER and HER. The negligible deactivations for water splitting were demonstrated in both 1M KOH for 12 hours and 10M KOH bulk alkaline for 20hours, though it is not sufficient since the long term water splitting under 1M KOH with 200 hours stability would be more acceptable, persuasive and comparable to those benchmark electrodes (Ref: doi: 10.1038/ncomms8261).
- 5.) Post 12-hour OER tests, the morphology of the NiFe/Ni-P electrodes were changed (Fig.S3 vs Fig.1b), why their performance remained intact during the stability tests?
- 6.) Different acronyms were used for the NiFe/Ni-P OER eletrodes, Ni₃Fe(OOH)₉ in abstract, NiFe-LDF in graphic abstract and NiFe in the full text.
- 7.) The OER catalyst was claimed as amorphous Ni₃Fe(OOH)₉ on the basis of TEM and SAED characterisations, which is problematic because the TEM and SAED are neither effective nor sufficient to determine the material composition and formula. The XPS cannot define the amount of -OH or hydrogen within NiFe catalysts either, in particular for LDH. Moreover, the TEM images were acquired from samples electrodeposited on FTO substrates rather than the Ni-P supported samples. The authors stated the FTO possessed different resistance to the Ni-P substrate, which therefore may lead to different crystallinity of the deposited catalysts. Further, HRTEM and SAED images were successfully applied to confirm the crystalline phase of the NiMo catalyst, hence the HRTEM image is necessary to confirm the amorphous phase of NiFe catalyst which would supports the SAED result too.

8.) The results of the Fig.5 (e) insets conflicted with results in Fig.5(f), weakening the argument for the exceptional stability of NiMo/Ni-P electrodes.

9.) The amount of O₂ and H₂ released from the control electrodes should have been compared with the highlighted electrodes in either full text or supporting information.

Therefore, the current quality of the manuscript doesn't fit the requirements of the prestigious Nature communications.

Modifications introduced in the Manuscript following the Reviewers' comments

Reviewer #1

This study reported an innovative way to transform various common substrates into flexible and efficient electrodes for water splitting. The results implied that cellulose paper and cotton have important advantages in many energy storage system regarding inexpensive, hierarchical porosity, intrinsic flexibility and tunable functionality when combined with other active materials. This work also demonstrated that high-performance water-splitting electrodes can be achieved by unique electrode configuration. Considering the importance of water splitting and the potential for practical applications, the manuscript could be published in Nature Common. after minor revisions as noted below:

Response: We thank the reviewer for appreciating this work.

01. The mass loading and weight ratio of Ni on the substrate are important factor for considering the conductivity and microstructure of the target electrodes, the authors should clearly point out these detailed values in the manuscript.

Response: Both the mass loading and weight ratio of Ni are included in the experimental section, on p. 24 of the revised manuscript.

02. The Ni nanoparticles on the cotton substrate had an average size of 32 nm which is larger than that of 23 nm on the cellulose paper substrate, and the corresponding Ni-cotton electrodes showed worse water splitting performance than Ni-cellulose paper. Is there any “nanosize-effect” in the water-splitting system?

Response: We are thankful to the reviewer for drawing our attention to this important aspect. Indeed the water-splitting performance is directly related to the exposed active sites of Ni nanoparticles which scale inversely with their size. Due to this nano-size effect, the Ni-paper electrodes with smaller 23 nm Ni nanoparticles are expected to have more exposed active sites as compared to the Ni-cotton substrates with larger 32 nm particles, which are also reflected from their relative catalytic performance. This point is highlighted in the revised manuscript on p. 19.

We further go on to establish a detailed understanding of the exceptional activities of Ni-paper electrodes. Through a series of new control experiments we demonstrate that two important structural features are unique to Ni-paper electrodes viz. i) nanostructuring of Ni and; ii) porosity of the underlying paper template, which are responsible for their improved activity over other commercial counterparts such as Ni-foam and Ni-foil. A detailed discussion describing this structure-function correlation is also included in the supporting information of revised manuscript in section S1.

03. The mass loadings of catalysts should be given rather than the mass.

Response: Mass loadings of the catalysts are now included in the experimental section on pp. 24-25 of the revised manuscript.

04. The authors should check the manuscript carefully since there has some format problems in the main text, for example, Page2, lin 31; Ref 13 and 27.

Response: The formatting errors are now corrected.

Reviewer #2

The authors report the synthetic method that leads to a novel current collector comprising Ni-modified paper. The material is well characterized and its further application as a support material for water splitting is also well characterized. However, I cannot recommend the publication of this paper in Nat. Commun. The reasons are listed as below:

Response: We thank the reviewer for appreciating our effort.

01. Flexibility of the electrode is not required by water splitting devices. Thus, one of the motivations of this work is not reasonable.

Response: We value the Reviewer's opinion that flexibility is trivial for water splitting devices; nonetheless we would like to draw attention to the fact that the individual half reactions such as OER are important from the view-point of flexible energy storage systems such as metal-air batteries. For example one can envisage that such light weight paper based electrodes which can be modified with bi-functional OER/ORR catalysts could be used as gas permeable cathodes in rechargeable Zn-air batteries for use in portable, flexible electronics [H. Dai et. al. *Nat. Commun.* **4**, 1805 (2013)]. Moreover, the ease with which the electrodes can be modified by surface treatments can also have important implications in other technologies such as flexible sensors. It is from this point-of-view that we feel the flexibility of electrodes can have many important advantages to researchers in allied areas and hence explicitly demonstrate the effect of flexibility on the performance of electrodes in this work.

In fact, the real motivation behind this research was to demonstrate how ubiquitous substrates which are cheap and easily accessible (such as paper and fabric) can be transformed into multifunctional electrodes using simple solution processable techniques. The demonstrated flexibility of the electrodes is considered to be one of the added advantages of such a value added transformation process and is not our only aim.

02. The Ni-P material is not advantageous over other commercially available collector materials, such as Ni foil, Ni foam and stainless steel. The preparation of Ni-P material is complex, with the use of Pd as the catalyst.

Response: We would like to mention that the improved electrochemical performance of Ni-P based electrodes towards both OER and HER as compared to the commercial Ni-foam was already demonstrated in the previous version of the manuscript. In order to gain a detailed understanding of the underpinning factors of such performance improvement, we have included new results from other control electrodes in the supporting information of revised manuscript in section S1. We successfully prove that two structural factors that are unique to Ni-P substrate i.e. nanostructured nature of the embedded Ni particles and porous framework of the underlying paper lend exceptional electrocatalytic activities to Ni-P as compared to other collector materials such as Ni-foil, Ni-foam etc. that lack either one or both of these structural paradigms. Thus, we not only demonstrate that Ni-P has advantageous

electrocatalytic activity but also present a structure-function correlation to understand the same.

The entire fabrication process of Ni-P electrodes consists of three simple dip coating steps which can be completed within 2 hours. The entire protocol is rather simple starting from the use of commonly available substrates such as paper or fabric and followed by a well-established solution processable deposition of Ni nanoparticles. Moreover our method is scalable since 30 electrodes of 1cm^2 area can be obtained within those 2 hours. Although, the method requires a dip coating step in PdCl_2 solution, the 15ml 0.6mM Pd bath can be reused 4-5 times before replenishing and thus produces ~ 150 electrodes. Hence it is apparent that the use of Pd as the catalyst is not a limiting factor for realizing the described process. In the opinion of the authors such a simple value added transformation of ubiquitous substrates through solution processing method is extremely cost effective and hence attractive.

03. Ni-Fe hydroxides and NiMo alloys are well established electrodes for OER and HER. So, the core content of this paper is lack of novelty.

Response: We would like to stress that the primary aim and motivation of this work is to demonstrate value added transformation of cheap, ubiquitous substrates such as common paper and cotton fabric into flexible current collectors for water splitting using a simple solution processed technique. The fact that NiFe hydroxides and NiMo alloys are well established for OER and HER was the very reason that these catalysts were chosen to coat the Ni-P substrates. Thus NiFe and NiMo are used in this work just as model representatives to demonstrate the concept of a paper-electrode-based alkaline water electrolyzer. As a result the purpose of this work is neither to develop new electrocatalysts, per-se, and nor to understand the mechanistic processes involved in their respective reactions, but to provide a proof-of-concept of an efficient alkaline electrolyzer formed out of ubiquitous substrates like paper/cloth through simple solution processing methods. In the opinion of the authors this is a novel endeavour as there have been no prior reports demonstrating such a capability.

Reviewer #3

The design and development of effective catalytic materials and corresponding OER/HER electrodes play profound roles in the electrocatalytic water splitting. The manuscript reported a combinatorial methodology for fabricating the flexible NiFe hydroxide/Ni-paper OER and NiMo/Ni-paper HER electrodes and demonstrated their exceptional performance for electrocatalytic water splitting in terms of their lower over-potentials and bulk stability. Through the typical characterisations, the authors claimed the excellent activities were due to the selected catalysts and the nanostructured porous surface of the electrodes which largely related to the Ni-paper substrates. The research topic well fits the scopes of the Natcomm.

The ideas of the combination of electroless plating of Ni on paper to form the Ni-Paper substrates and using the obtained substrates to electroplate the state-of-the-art catalysts are smart, though the methodology is not novel. The context is not concise and significant weakness exists in the manuscript as detailed following.

Response: We thank the reviewer for appreciating this work.

01. The paper addressed the flexibility and the exceptional activity of the Ni/paper substrates for OER and HER electrodes, yet the understanding of the underpinning improvements of OER and HER reaction was not confidently attempted nor persuasive.

Response: We thank the reviewer for this helpful comment. In the previous version of the manuscript a preliminary understanding of the impressive electrocatalytic activity of Ni-paper based substrates was obtained by comparing their performances with those of commercial Ni-foam. Interestingly, it was observed that the Ni-paper substrates are intrinsically much more active than Ni-foam towards both OER and HER under otherwise identical conditions. Ni-foam although being a highly porous substrate is composed of bulk Ni metal which is in stark contrast with the nanoparticulate Ni of Ni-paper substrate (average size of 23nm as observed from electron microscopy). Thus, the improved activity of Ni-paper was traced back to the nanostructured nature of its interface.

To further our understanding of the underpinning factors and develop a structure-function correlation for Ni-paper electrodes, we now include the results of a comparative study with 2 more control electrodes. To begin with, we hypothesize that besides the nanostructured nature of Ni, the intrinsic porosity imparted to the electrode by the underlying cellulose paper template would also be a contributing factor for the observed activities of Ni-paper. To test this hypothesis a nanostructured but non-porous control electrode was prepared by electroless plating of Ni nanoparticle thin film on FTO-glass (abbreviated as Ni-FTO) using a process identical to that used for paper substrates. The electrocatalytic activity of Ni-FTO towards OER test reaction was found to be much lower than Ni-paper, thereby validating our hypothesis of the importance of porous interfaces.

Finally, this was confirmed by the negligible activity observed for Ni-foils, which are neither porous and nor nanostructured. To sum it all, we conclude that two structural features that are unique to Ni-paper electrodes, namely i) nanostructured nature of Ni and; ii) porous nature of the underlying paper substrate act in a synergistic fashion to lend exceptional electrocatalytic activities to Ni-paper electrodes. While Ni-nanoparticles with large surface-to-volume ratio have abundance of exposed active sites for catalysis, the interweaving pores can render faster mass-transport kinetics. Thus by a careful choice of control electrodes, this comparative study establishes the underpinning structure-function correlation in Ni-paper electrodes. These new results and the corresponding discussion are added to the Supporting Information of the revised manuscript in section S1 and Figure S3.

02. In the experimental section, the important parameters, the sizes of the Ni-P substrates, were not presented for OER and HER electrodes.

Response: The sizes of Ni-P electrodes are now specified for all electrocatalysis experiments in the revised manuscript on p. 26.

03. The comparisons of the overpotential based on the 10mA/sq or 50 mA/sq currents were conducted either in different electrolytes or different electrode sizes (tables in the context and supporting information), which shakes the foundation of the claims to exceptional performance of the flexible electrodes.

Response: For the data presented in Table 1 and Table 2 the overpotentials are measured for currents per unit area of the respective electrodes which thereby normalizes the varying size effect for different systems compared therein. All the overpotentials required for reaching 50mA/cm² for OER and 10mA/cm² for HER presented in this work were measured under identical conditions in a 1M KOH electrolyte. However, as pointed out by the Reviewer,

Tables S1 and S2 presented for a comparative literature survey have few entries based on 0.1 M KOH which is different from the conditions used in this work. These entries are now removed in the revised version of the manuscript for a more rigorous comparison. We would like to point out that the electrocatalytic activity of Ni-paper electrodes is much improved over earlier reports under otherwise identical conditions.

04. Stability of the electrodes is the pivotal property for OER and HER. The negligible deactivations for water splitting were demonstrated in both 1M KOH for 12 hours and 10M KOH bulk alkaline for 20hours, though it is not sufficient since the long term water splitting under 1M KOH with 200 hours stability would be more acceptable, persuasive and comparable to those benchmark electrodes (Ref: doi:10.1038/ncomms8261).

Response: We agree with the reviewers that more stringent and long term stability experiments for overall water splitting using paper electrodes can further help in establishing their exceptional activities. In the revised version of the manuscript we now include results of electrochemical stability towards overall water splitting of NiFe/Ni-P and NiMo-Ni-P electrodes in excess of 200 hours (Figure 8f). As observed from the chronoamperometric traces for total water splitting in 1M KOH, the paper electrodes show exceptional stability over a period of close to 10 days without any significant attenuation of the output current. Additionally; similar long term stability tests were also performed for bare Ni-P electrodes for more than 200 hours (Figure S10) which also reveal their impressive durable nature. These exhaustive tests provide a rigorous proof for the exceptional stability and impressive robustness of Ni-paper based electrodes, besides those already demonstrated under harsh environments such as 10M KOH. The results from these stability tests are discussed on p. 21 of revised version of the manuscript. We thank the reviewer for improving the quality of this work through this suggestion.

05. Post 12-hour OER tests, the morphology of the NiFe/Ni-P electrodes were changed (Fig.S3 vs Fig.1b), why their performance remained intact during the stability tests?

Response: The reviewer raises an important and valid point about performance of OER electrodes after 12h stability tests and the accompanying morphology change. From the FE-SEM micrographs in Figure S4 it is quite apparent that the initial vertically aligned flake type morphology of NiFe gives way to a rice-grain type structure after 12h electrolysis. Surprisingly enough this morphology transformation has a minimal effect on catalytic activity of NiFe/Ni-P as observed from similar η^{50} values before and after stability tests (inset Figure 4e, main text). Moreover, the stable current outputs delivered by NiFe/Ni-P in a two-electrode device for more than 200h of continuous operation (Figure 8f, main text) also suggest sustained activity post morphology alteration. We would like to state that similar results were obtained on repeated trials at varying current densities for 12h chronoamperometric tests. Careful analysis of FE-SEM images post stability tests, however, reveals presence of a porous microstructure for NiFe. To investigate this aspect further, the electrochemically active surface area (ECSA) of NiFe/Ni-P electrode was compared before and after 12h stability tests. The similar slopes obtained for the plot of current density v/s scan rate (Figure S5) indeed demonstrates that the ECSA of NiFe/Ni-P post stability test is almost identical to that before. This helps us to rationalize the observations. We can conclude that although the morphology of NiFe is physically transformed over the course of long term electrolysis, the active area that contributes to the electrochemical current still remains the same and hence is responsible for stable electrode performance. This clarification is now

added in the revised version of the manuscript on p. 13 and section S3 of supporting information.

We thank the reviewer for improving the quality of this work through this comment.

06. Different acronyms were used for the NiFe/Ni-P OER electrodes, Ni₃Fe(OOH)₉ in abstract, NiFe-LDF in graphic abstract and NiFe in the full text.

Response: This anomaly is corrected and a single acronym of NiFe is used to describe the OER catalyst and NiFe/Ni-P for the electrode in the abstract, graphic abstract and full text.

07. The OER catalyst was claimed as amorphous Ni₃Fe(OOH)₉ on the basis of TEM and SAED characterisations, which is problematic because the TEM and SAED are neither effective nor sufficient to determine the material composition and formula. The XPS cannot define the amount of –OH or hydrogen within NiFe catalysts either, in particular for LDH. Moreover, the TEM images were acquired from samples electrodeposited on FTO substrates rather than the Ni-P supported samples. The authors stated the FTO possessed different resistance to the Ni-P substrate, which therefore may lead to different crystallinity of the deposited catalysts. Further, HRTEM and SAED images were successfully applied to confirm the crystalline phase of the NiMo catalyst, hence the HRTEM image is necessary to confirm the amorphous phase of NiFe catalyst which would supports the SAED result too.

Response: The TEM and SAED characterizations were performed to establish the amorphous nature of NiFe catalyst while XPS was performed to understand the composition. Although we agree with the reviewer about the discrepancy in quantification of OH or hydrogen using XPS, we would like to draw attention to similar assignment made by Lu *et al.* [*Nat. Commun.* **6**, 6616 (2015)]; from where the methodology of depositing NiFe catalyst was adapted. However, to maintain rigour in the structural characterization of the catalyst we now use the composition of Ni₃Fe(OOH)_x in the revised manuscript.

As mentioned in the manuscript, it is extremely difficult to isolate the NiFe or NiMo catalysts for HR-TEM characterization from the Ni-paper substrate without avoiding any contamination from the cellulose strands due to surface scratching. For this purpose, these catalysts were deposited onto FTO substrates from which the catalyst can be easily dislodged by sonication or gentle scratching. FTO was chosen because of its inert yet conductive nature which is essential for electrodeposition. We would like to stress that the crystallinity of NiMo is not altered as observed from HR-TEM, SAED and XRD characterizations which perfectly match prior literature reports on electrodeposited NiMo. As suggested by the reviewer HR-TEM and SAED characterization of NiFe is now added in the revised manuscript (Figure 1d), both of which confirm its amorphous nature. Thus although a crystalline FTO substrate is used to deposit NiFe for electron microscopy studies, its amorphous nature is still intact which also suggests that FTO does not alter the structural identity of the electrodeposited catalysts.

08. The results of the Fig.5 (e) insets conflicted with results in Fig.5(f), weakening the argument for the exceptional stability of NiMo/Ni-P electrodes.

Response: We would like to point out that the inset in Figure 5e presents the LSV curves of NiMo/Ni-P catalyst after a steady state stability test of 12h. The overpotential at 10mA/cm² before and after 12h was found to be very similar whereas a small increase at higher overpotentials was observed resulting from an increase in mass transport resistance due to

vigorous gas bubbling. On the other hand, the results in Figure 5f show the accelerated degradation tests (non-steady state conditions) recorded after 2000 CV cycles for NiMo/Ni-P. The overpotential at $10\text{mA}/\text{cm}^2$ before and after 2000 CV cycles was found to be almost identical, thereby supporting the observation made for chronoamperometric stability experiments.

09. The amount of O₂ and H₂ released from the control electrodes should have been compared with the highlighted electrodes in either full text or supporting information.

Response: The results of Faradaic efficiency measurement for control electrodes of bare Ni-P and their comparison with the highlighted electrodes is now discussed in section S4, supporting information.

Reviewers' comments:

Reviewer #1 (Remarks to the Author):

The authors have well-addressed the reviewers' comments, I think it is suitable to be accepted after adding some comments to further support this study.

The improved electroactivity of the flexible electrodes can also owing to the unique features for paper or cotton substrate: 1, it serves as an interior electrolyte reservoir to provide additional pathway for ion transport; 2, the flexible substrate renders reduced internal strain during repeatedly electrochemical process (Adv. Energy Mater. 2017, 10.1002/aenm.201700130; ACS Nano 2013, 7, 6037).

Reviewer #2 (Remarks to the Author):

I still do not recommend the publication of this paper in Nature Communications.

1. About flexibility:

1) I want to emphasize again that flexibility of the electrode is not highly demanded by water splitting devices. Flexibility is also not the bottleneck of the development of water splitting devices. The authors also agree with my comments in their response.

2) Although I agree with the authors' response that wearable Zn-O₂ batteries require flexible electrodes, but I want to bring another fact to the authors' attention. A good OER catalyst for water splitting does not mean that it must be a good OER catalyst for Zn-O₂ batteries. Different electrodes are required for the two types of devices. If the authors continue to claim the meaning of flexible electrodes, construction of a wearable Zn-O₂ battery might be the suitable choice.

3) The authors describe a lot of importance of flexibility in the introduction, but inflexibility is not the core problem of water splitting devices. Thus, the authors might mislead authors.

2. About the synthetic method

The authors claim that they report a extremely cost effective method for fabricating conductive electrode because of the use of cheap and easily accessible substrate, such as paper. However, I want to bring an important fact for the authors: during the synthesis, noble metal Pd are used despite of its low amount. How to know this method is cost effective?

Thus, the cost-effectiveness is overselling in this paper. The real motivation is an untenable argument.

3. About water splitting catalysts

Ni-Fe hydroxides and NiMo alloys are widely used as effective electrocatalysts for OER and HER. Although good catalytic activities are also obtained in this paper, the novelty is not high.

Reviewer #3 (Remarks to the Author):

The authors properly addressed most of the questions raised previously, though the paper is still quite lengthy as a communication. Some minor scientific questions are aware:

- 1.) Is the Pd catalysts still laid down on or within the Ni-paper after electroless deposition? What is their fate? If they still exist on the paper, would they contribute to the conductivity and the activity?
- 2.) Taking into the multiple interfaces existing on the electrodes, the equivalent circuit employed cannot properly reflect the electrodes structure in interpreting R_{ct} .
- 3.) Has the paper substrate been damaged when using in the corrosive environment? In particular 10M KOH conditions.
- 4.) Can the authors calculate reaction Turnover frequencies (ToF)?
- 5.) A lot of experimental data were presented without considering the experimental standard errors.
- 6.) In the reply to previous Question 5, the authors argued the grown particles with different morphology did not affect the ECSA, while the excellent performance of the electrodes were attributed to the nano-size effects. This reply is either confusing or conflicting with the main finding of the paper.
- 7.) The selected areas for electron diffraction in TEM and zone for EDAX in SEM should have been marked.

I would like to recommend to accept the manuscript for publication, though further revision is necessary in terms of the current quality.

Modifications introduced in the Manuscript following the Reviewers' comments

Reviewer #1

The improved electroactivity of the flexible electrodes can also owing to the unique features for paper or cotton substrate: 1, it serves as an interior electrolyte reservoir to provide additional pathway for ion transport; 2, the flexible substrate renders reduced internal strain during repeatedly electrochemical process (Adv. Energy Mater. 2017, 10.1002/aenm.201700130; ACS Nano 2013, 7, 6037).

Response: We thank the reviewer for contextualizing the role of electrode flexibility and for additional references. The unique features of flexible electrodes are elaborated more in the main text of the revised manuscript on p. 3 with the suggested references 23 and 24.

Reviewer #2

1. About flexibility:

1) I want to emphasize again that flexibility of the electrode is not highly demanded by water splitting devices. Flexibility is also not the bottleneck of the development of water splitting devices. The authors also agree with my comments in their response.

Response: With due consideration to the Reviewer's concern herein we mention few recent studies which show that strain engineering of electroactive materials through controlled manipulation of mechanical flexibility plays a decisive role in interfacial electrochemical processes that directly affect the device performance. Particularly Wang et. al (*Science* **354**, 1031 (2016)) have demonstrated how strain tuning of electrocatalysts improves their ORR catalytic activity while Zhang et. al (*PNAS* **114**, E11082 (2017)) explored the effects of mechanical flexibility for strain controllable charge storage. Similarly, other studies have shown that flexible substrates render reduced internal strain during repeated electrochemical processes (*Adv. Energy Mater.* **7**, 1700130 (2017); *ACS Nano* **7**, 6037 (2013)). Thus in the light of these studies it is quite reasonable to expect that electrode flexibility, although not the bottleneck for water splitting devices, can certainly influence their electrochemical activity through local strain fields and hence cannot be ignored. The role of flexible substrates and emergent strain fields for tuning electrocatalytic activity of heterogeneous catalysts is currently being pursued in our group.

Moreover, as argued in our previous response to the same comment, flexible OER/ORR electrocatalysts are particularly attractive for wearable Zn-air batteries. Thus, although not a bottleneck for water splitting devices, flexible electrodes for individual half reactions of water splitting are technologically relevant. We indeed have demonstrated successful fabrication and decent performance of such a paper based Zn-air battery. Although currently this work is under progress we would like to share with you some preliminary results to demonstrate that such a capability is indeed possible (please see answer to next comment).

2) Although I agree with the authors' response that wearable Zn-O₂ batteries require flexible electrodes, but I want to bring another fact to the authors' attention. A good OER catalyst for water splitting does not mean that it must be a good OER catalyst for Zn-O₂ batteries. Different electrodes are required for the two types of devices. If the

authors continue to claim the meaning of flexible electrodes, construction of a wearable Zn-O₂ battery might be the suitable choice.

Response: Although the construction of a wearable paper-based Zn-air battery is an altogether new research problem that we are currently investigating, we would like to share with the reviewer some of the promising early results of this endeavour. However, we would like to note that these results are beyond the scope of the current manuscript and hence included in the supporting information, Figure S13.

Zn-air battery fabrication: The fabrication of the wearable Zn-air battery is demonstrated in the schematic. A layer-by-layer assembly was employed with Zn foil (thickness 0.25 mm, Alfa Aesar) as anode, PVA gel film soaked in 6M KOH as electrolyte and Ni-P coated with Co₃O₄ catalyst ink as gas diffusion electrode (GDE). To generate absolute contact between the layers, this assembly was mechanically pressed for 30min. Over an active area of 1cm × 1cm, the catalyst loading at GDE was 1.02mg.

Battery Performance: Battery performance was analyzed with a constant load charging and discharging process. An impressive specific capacity of 1003 mAh/g was obtained for our solid state battery at a modest discharging current of 0.2mA/cm², which delivered a steady state output voltage of 0.9V (Figure R1a). The effect of device flexibility on its performance was also studied by charging and discharging the battery in a bent state at 0.5mA/cm² (Figure R1b). Expectedly even under mechanical stress the battery delivered a specific capacity of 700 mAh/g, which corresponds to ~97% of the specific capacity value obtained in planar cell (726 mAh/g). Further work is underway to test battery performance in a controlled and variable mechanical loading. To show the practicality of this device we also demonstrate that 2 Zn-air batteries connected in series successfully light-up a commercial LED (Fig R1c,d). These preliminary un-optimized results are indeed very promising and further work is currently underway for systematic improvements.

We hope that, through these preliminary results on the use of Ni-paper electrodes for flexible Zn-air batteries, we were able to convince the reviewer on the high potentiality of the Ni-paper as flexible and versatile substrate for allied technological applications.

3) The authors describe a lot of importance of flexibility in the introduction, but inflexibility is not the core problem of water splitting devices. Thus, the authors might mislead authors.

Response: In light of the importance of mechanical flexibility for tuning electrochemical activity as well as the technological utility of individual half reactions for flexible, wearable batteries we have modified the section on electrode flexibility. So as not to mislead the potential readers of this work we add in the Introduction section of the revised manuscript the importance of electrode flexibility in the appropriate context of the above mentioned advances and benefits. We also envisage that a clear demonstration of flexible nature of electrodes in this work can serve as an important platform for other researchers, particularly in battery and sensors community, to fabricate functional devices out of commonly available substrates.

2. About the synthetic method

The authors claim that they report a extremely cost effective method for fabricating conductive electrode because of the use of cheap and easily accessible substrate, such as

paper. However, I want to bring an important fact for the authors: during the synthesis, noble metal Pd are used despite of its low amount. How to know this method is cost effective?

Thus, the cost-effectiveness is overselling in this paper. The real motivation is an untenable argument.!

Response: As pointed out by the reviewer, the cost effectiveness of our approach arises mainly from the use of cheap resources such as paper and cloth which are abundantly and easily accessible. Re-iterating our previous response, the use of Pd as the catalyst is not a limiting factor for realizing the described process since the cost of Pd sensitization is just ~30% of the total cost of the entire process which is ~1 USD, roughly 80 times less (even by a conservative estimate) than that of commercially available Ni foam from Sigma Aldrich that costs ~79 USD for 10cm × 10cm sheet of a flexible electrode. Moreover the minimal quantity of Pd in the dilute sensitization bath can be reused over multiple rounds before replenishing.

In order to explicitly show the cost effectiveness of our approach we present below a preliminary estimate of average cost incurred for preparing 1 cm² of the flexible Ni-paper.

Sensitizing solution (0.05M of SnCl₂.2H₂O)

SnCl₂.2H₂O (INR 2028, 250g MERCK)

Amount required per sensitization bath = 0.45g (40mL)

Cost per bath – INR 3.65

Each bath can be used to sensitize at least 4 stripes (6cm × 2cm) of paper.

Price per Ni paper stripe = INR 0.91

Activating solution (100ug/mL PdCl₂)

PdCl₂ (INR 9310, 2g Alfa Aesar)

Amount required per bath = 4mg (40mL)

Price per bath = INR 18.62

Each bath can be used to activate at least 10 stripes (6cm × 2cm) of paper.

Price per Ni paper stripe = INR 1.86

Ni Plating Solution (30mL)

I. NiCl₂.6H₂O (INR 2388, 500g MERCK)

Amount required per bath = 1.68 g

Price per bath = INR 8.02

II. Na₃C₆H₅O₇.2H₂O (INR 288, 500g MERCK)

Amount required per solution = 2.5g

Price per bath = INR 1.44

III. NH₄Cl (INR 234, 500g MERCK)

Amount required per solution = 1.68 g

Price per bath = INR 0.79

IV. NaH₂PO₄.2H₂O (INR 644.44 , 500g MERCK)

Amount required per bath = 0.44 g

Price per bath = INR 0.57

Total price per bath = INR 10.82

Each bath can be used to plate 2 stripes (6cm × 2cm) of paper.

Price per Ni paper stripe = INR 5.41

Total price per stripe (6cm × 2cm) = INR 8.18
= INR 0.68/cm² ≈ 0.01 USD/cm²

Thus our method allows production of a 10cm × 10cm flexible electrode at a cost of ~1 USD and we would also like to add that this estimated cost is an upper bound and further cost reduction per cm² of Ni-paper is possible through careful optimization. In the opinion of the authors such a simple, value added transformation of ubiquitous substrates into functional electrodes through solution processing methods is extremely cost effective and hence attractive.

3. About water splitting catalysts

Ni-Fe hydroxides and NiMo alloys are widely used as effective electrocatalysts for OER and HER. Although good catalytic activities are also obtained in this paper, the novelty is not high.

Response: As explained in our previous response to the same comment, the primary motivation of this work is to demonstrate the capability of value added transformation of cheap and ubiquitous resources into functional electrodes for water splitting. We would like to argue that such a capability has not been demonstrated before in the literature which makes our contribution novel and particularly relevant for resource poor settings. We would like to reiterate that the electrocatalysts chosen for this study are model representatives to simply build a proof-of-concept device. Moreover, we have also successfully demonstrated the utility of such electrodes in other important technologies like flexible Zn-air paper-batteries (see Response to comment 1 part 3). To conclude, the purpose of this work is neither to develop new electrocatalysts, per-se, and nor to understand the mechanistic processes involved in their respective reactions, but to provide a proof-of-concept of an efficient alkaline electrolyzer formed out of ubiquitous substrates like paper/cloth through simple solution processing methods. In the opinion of the authors this is a novel endeavour as there have been no prior reports demonstrating such a capability.

Reviewer #3

1.) Is the Pd catalysts still laid down on or within the Ni-paper after electroless deposition? What is their fate? If they still exist on the paper, would they contribute to the conductivity and the activity?

Response: As discussed previously in the main text, Pd was not detected from the full-scan XPS spectrum of Ni-paper, NiFe/Ni-P and NiMo-Ni-P electrodes. However, since XPS is only a surface technique, it may not be able to detect presence of trace metals trapped within the volume of the porous paper substrate. To answer this question, we resorted to cross-sectional EDAX analysis of bare Ni-paper electrodes. As expected from the stepwise protocol of electroless Ni-plating, presence of trace amounts of Pd (~1.2 %) and Sn (~2.6 %) was detected within the Ni-paper cross-section (Fig. S1a). We note that due to the porous nature of paper electrodes, the electrolyte can still access the buried metal impurities. Hence we further investigate whether Pd or Sn play any role in the observed electrocatalytic activity and/or conductivity of the electrodes. To that end, we recorded the two-probe I-V response of

Pd and Sn activated paper which is a straight line with slope similar to that of bare cellulose paper (Fig. S1b) showing that the Pd and Sn activated paper is as resistive as bare cellulose paper. Moreover, from the LSVs of the Pd and Sn activated paper under both OER and HER conditions, no catalytic currents were observed even under overpotentials in excess of 1 V for both the reactions which is almost identical to the response of a bare cellulose paper (Fig 4a and 5a). This evidently attests to the non-catalytic nature of Pd and Sn towards alkaline electrolysis of water (Fig. S1c). These observations clearly suggest that the trace amounts of Pd and Sn do not alter in any significant way either the conductivity or the electrocatalytic activity of the electrodes. Thus we conclude that Pd and Sn used during electroless plating process are mere “spectator” species. The text is modified on p. 5, 6, 11 and 14-15.

2.) Taking into the multiple interfaces existing on the electrodes, the equivalent circuit employed cannot properly reflect the electrodes structure in interpreting Rct.

Response: We agree with the reviewers comment that a Randle’s equivalent circuit does not appropriately model the response of porous electrodes as it effectively captures a single electrochemical interface. To capture the multiple electrochemical interfaces emerging from a paper electrode we now use a transmission line model to fit the EIS spectrum. The transmission line equivalent circuit, which is essentially an infinite network of uniformly distributed RC elements that correspond to the large number of electrochemical interfaces, is known to be a better model for porous electrodes and hence is the more appropriate choice for this work. The fitted data and the circuit diagram are presented in Table 1 and 2 and the insets of Figure 4d, 5d, respectively, in the revised manuscript.

3.) Has the paper substrate been damaged when using in the corrosive environment? In particular 10M KOH conditions.

Response: The paper substrate remains unaltered even after 12h electrolysis in 10M KOH without any physical disintegration. FESEM micrographs, provided in the supporting information of the revised manuscript (Fig. S12) for both the anode and the cathode after continuous 12h electrolysis in 10M KOH clearly show the presence of intact catalyst layer for both NiFe/Ni-P anode and the NiMo/Ni-P cathode. Digital images of the anode and the cathode after bulk electrolysis in corrosive environment are also provided in Fig. S12.

4.) Can the authors calculate reaction Turnover frequencies (ToF)?

Response: The reaction turnover frequencies (ToF) are now provided on p. 11, 14 of the revised manuscript for different catalytic electrodes. In particular ToF for NiFe/Ni-P was calculated to be 0.13 s^{-1} which is an order of magnitude higher than that of unmodified Ni-P ($\text{ToF} = 0.015 \text{ s}^{-1}$) at an overpotential of 250mV. Similarly the ToF for NiMo/Ni-P (0.33 s^{-1}) was significantly higher than that of Ni-P (0.024 s^{-1}) towards HER at overpotential of 130mV. We would like to mention that these ToF values are calculated using the assumption that all redox active Ni sites contribute to catalytic current observed in the LSV. As such, the reported values are a lower bound on the ToF. The method for estimating ToF is provided in the experimental section of the manuscript on p. 28. Briefly we use the method established by Yeo and Bell (*J. Phys. Chem. C* **116**, 8394 (2012)) according to which electrochemical equivalent ToF is given by

$$\text{ToF} = \frac{i_{\text{Ni}}}{n \times F \times N_{\text{Ni}}}$$

where i is the current, N_{atoms} is the number of atoms or active sites, n is the number of electrons transferred, F is the Faraday's constant and N_A is Avogadro's number. Here N_{atoms} is estimated from the charge storage capacity of the main Ni(II)/Ni(III) redox peak.

5.) A lot of experimental data were presented without considering the experimental standard errors.

Response: Standard deviation based on $n = 3$ electrodes are now provided for the relevant electrochemical data in the revised manuscript especially in Tables 1 and 2.

6.) in the reply to previous Question 5, the authors argued the grown particles with different morphology did not affect the ECSA, while the excellent performance of the electrodes were attributed to the nanosize effects. This reply is either confusing or conflicting with the main finding of the paper.

Response: In the reply to Question 5 of previous report we explained that even though the NiFe catalyst layer of NiFe/Ni-P electrodes changes its morphology from an initial flake type structure to rice-grain type morphology at the end of 12h bulk electrolysis, the resulting electrochemically active surface area remains the same. This was demonstrated from the nearly identical slopes obtained for peak current density v/s scan rate plots. Thus, the electrochemically relevant surface area does not change even if the morphology of the catalyst layer changes. This is also evident from the stable long-term current observed for these electrodes in excess of 200h.

On the other hand the nano-size effect was described in reference to the bare Ni-paper electrodes (i.e without the catalyst layer). It was demonstrated that the unmodified Ni-paper has an intrinsic electrocatalytic activity that emerges from the nanostructured Ni particles which is in stark contrast to other conventional current collectors such as Ni-foam or Ni-foil that offer a bulk, non-porous interface and are inert. The nanostructured nature of unmodified Ni-paper current collector is preserved even after 12h electrolysis as seen from the FESEM micrographs in Fig S5 b-d.

Thus the two phenomena are unrelated and not in conflict with the main findings of this work.

7.) The selected areas for electron diffraction in TEM and zone for EDAX in SEM should have been marked.

Response: Done in Fig. 1 and 3.

Reviewers' comments:

Reviewer #1 (Remarks to the Author):

From my point of view, the manuscript has been revised accordingly, all the comments raised by previous reviewers have been well-addressed. So, I recommend it to be accepted without further modification.

Reviewer #2 (Remarks to the Author):

Although the authors have made many efforts to improve the manuscript and to try to reverse our opinions on this manuscript. However, I still can not recommend the publication of this manuscript.

1) The authors have agree with my comment that flexibility is not the necessary of water splitting electrodes. So, the discussion on flexibility in the introduction should be removed.

2) The flexibility of the material should be quantified.

3) The authors compared the cost of the material with that of nickel foam. How about the cost comparison between the material and nickel foil? How about the cost comparison between the material and the nickel foil/foam that is not from Aldrich-Sigma?

4) The method used by the authors for the fabrication of nickel film is transitionally called as "Electroless Nickel Plating". This method was first developed in 1946. Please see literature: "A. Brenner, G.E. Riddell, J. Res. Nat'l Bur. Std. 37 (1946) 31. A. Brenner and G. E. Riddell, U.S. Pat. 2,582,283 (1950)." Now this method has been commercialized for nickel plating. A surface coated in electroless nickel has even be used on non-conductive (or non-metallic) surfaces which allows for plating of a wider variety of base materials.

Overall, the novelty and significance of this work are not high enough to guarantee the publication in NC.

Reviewer #3 (Remarks to the Author):

The authors have provided new and solid data to support their arguements. Although the manuscript is still quite lengthy as a communication paper, I am satified with data presented and the idea delivered in the paper. Hence, I would like to recommend it to be accepted for publication.

Additional literature context

References 26, 27 and 28 are added in the revised manuscript to provide literature context on the electroless nickel plating technique and its use to coat common, carbon substrates similar to those studied in this manuscript.

Modifications introduced in the Manuscript following the Reviewers' comments

Reviewer #1

1) From my point of view, the manuscript has been revised accordingly, all the comments raised by previous reviewers have been well-addressed. So, I recommend it to be accepted without further modification.

Response: We thank the reviewer for helpful comments along the way in this process which has helped us to improve the quality of work.

Reviewer #2

1) The authors have agree with my comment that flexibility is not the necessary of water splitting electrodes. So, the discussion on flexibility in the introduction should be removed.

Response: From the successful fabrication and demonstration of a paper-based rechargeable Zn-Air battery in the last version of the revised manuscript, we validated our argument that electrode flexibility, although not a straight-forward requirement for water splitting, is important in the individual electrochemical half reactions for allied technologies that draw directly from reactions of water splitting and hence needs to be explicitly demonstrated for the broader benefit of the research community.

The discussion on flexibility in the second paragraph of the introduction section stems from a literature review of the paper electrodes and paper electronics developed by other researchers. Since this work is on paper electrodes, we feel this brief review of previous research is crucial to maintain continuity in our discussion. The other paragraphs of the Introduction section deal with transforming ubiquitous substrates and an initial overview of our work.

2) The flexibility of the material should be quantified.

Response: Flexibility has been quantifiably demonstrated in terms of bending angle of the paper electrodes as already shown in Figure 7b; wherein the electrochemical performance did not alter with bending up to 180°. Furthermore the stiffness of the electrodes is estimated based on the method developed by F. T. Carson and V. Worthington [*J. Res. Natl. Bur. Stand.* **49**, 385 (1952)] in which the average stiffness of paper has been shown to range 2-12 g-cm. In an analogous manner we calculate the bending stiffness (flexural rigidity) of a 2 × 6 cm piece of electrode to be 4.4 g-cm at a bending angle of 30°. This value is now included on page 6 of the revised manuscript along with the above reference (Ref 33).

3) The authors compared the cost of the material with that of nickel foam. How about the cost comparison between the material and nickel foil? How about the cost comparison between the material and the nickel foil/foam that is not from Aldrich-Sigma?

Response: As described previously in the preliminary cost estimation, the average cost incurred for Ni-paper electrodes in this work is ~1 USD for a 100cm² sheet. This is considerably cheaper than different commercial counterparts such as Ni-foam (Sigma) at ~79 USD, Ni-foil (Sigma, 0.5 mm) 313 USD, Ni foam (Goodfellow) 61 USD, Ni-foil (Goodfellow, 0.025 to 2 mm) ~ 800 to 2500 USD, Ni-foil (Alfa Aesar, 0.5 mm) 806 USD (Note: All prices quoted for 100 cm² area).

We would also like to add that this estimated cost for electrodes presented in this work is an upper bound and further cost reduction per cm² of Ni-paper is possible through careful optimization.

4) The method used by the authors for the fabrication of nickel film is transitionally called as "Electroless Nickel Plating". This method was first developed in 1946. Please see literature: "A. Brenner, G.E. Riddell, J. Res. Nat'l Bur. Std. 37 (1946) 31. A. Brenner and G. E. Riddell, U.S. Pat. 2,582,283 (1950)." Now this method has been commercialized for nickel plating. A surface coated in electroless nickel has even been used on non-conductive (or non-metallic) surfaces which allows for plating of a wider variety of base materials.

Response: As pointed out by the reviewer, electroless plating is a robust and well-established process which has been traditionally employed in metallurgy for anti-corrosion coatings. Recently, this method has received renewed attention essentially because of its easy-to-implement solution processed nature and scalability as detailed in references 26 to 28 (and references within those articles/reviews). Much of the work in this regard, hitherto, has focused to demonstrate the utility of this method to create conducting patterns on polymer substrates, deposit passivation layers/surface treatments or to develop passive circuit elements such as interconnects, resistors and radio-frequency antennae. However, in our present work we demonstrate a new application in the form of fully functional active catalytic electrodes. We leverage upon the above mentioned two advantages of this method (viz. scalability and solution processing) to transform ubiquitous substrates into functional electrocatalytic systems, which in our opinion is a novel endeavour with no prior precedent. Moreover, we would also like to draw your attention to the versatility of our approach that has allowed us to extend the application space of our materials and processes to the domains of rechargeable batteries and sensors which we are actively pursuing further in our group. The suggested journal reference (Ref 25) is added in the revised manuscript along with additional literature context (Ref 26 to 28) in the revised manuscript.

Reviewer #3

1) The authors have provided new and solid data to support their arguments. Although the manuscript is still quite lengthy as a communication paper, I am satisfied with data presented and the idea delivered in the paper. Hence, I would like to recommend it to be accepted for publication.

Response: We thank the reviewer for helpful comments along the way in this process, particularly in the context of material characterization and suggestion about long term electrochemical activity, which has helped us to improve the quality of this work.

REVIEWERS' COMMENTS:

Reviewer #2 (Remarks to the Author):

The authors have addressed my concerns. I recommend the publication of this paper.

Reviewer #2

1) The authors have addressed my concerns. I recommend the publication of this paper.

Response: We thank the reviewer for helpful comments along the way in this process which has helped us to improve the quality of work.